# Morse: Dual-Sampling for Lossless Acceleration of Diffusion Models

**Chao Li** [1]  **Jiawei Fan** [1]  **Anbang Yao** [1]

## Abstract

In this paper, we present *Morse*, a simple dual-sampling framework for accelerating diffusion models losslessly. The key insight of Morse is to reformulate the iterative generation (from noise to data) process via taking advantage of fast jump sampling and adaptive residual feedback strategies. Specifically, Morse involves two models called *Dash* and *Dot* that interact with each other. The Dash model is just the pre-trained diffusion model of any type, but operates in a jump sampling regime, creating sufficient space for sampling efficiency improvement. The Dot model is significantly faster than the Dash model, which is learnt to generate residual feedback conditioned on the observations at the current jump sampling point on the trajectory of the Dash model, lifting the noise estimate to easily match the next-step estimate of the Dash model without jump sampling. By chaining the outputs of the Dash and Dot models run in a time-interleaved fashion, Morse exhibits the merit of flexibly attaining desired image generation performance while improving overall runtime efficiency. With our proposed weight sharing strategy between the Dash and Dot models, Morse is efficient for training and inference. Our method shows a lossless speedup of $1.78\times$ to $3.31\times$ on average over a wide range of sampling step budgets relative to 9 baseline diffusion models on 6 image generation tasks. Furthermore, we show that our method can be also generalized to improve the Latent Consistency Model (LCM-SDXL, which is already accelerated with consistency distillation technique) tailored for few-step text-to-image synthesis. The code and models are available at https://github.com/deep-optimization/Morse.

[1]Intel Labs China. Correspondence to: Anbang Yao <anbang.yao@intel.com>.

*Proceedings of the 42nd International Conference on Machine Learning*, Vancouver, Canada. PMLR 267, 2025. Copyright 2025 by the author(s).

## 1. Introduction

Diffusion models (DMs), a class of likelihood-based generative models, have achieved remarkable performance on a variety of generative modeling tasks such as image generation (Ho et al., 2022), text-to-image generation (Zhang et al., 2023), video creation (Blattmann et al., 2023), text-to-3D synthesis (Poole et al., 2023) and audio synthesis (Liu et al., 2022). The powerful generalization ability of DMs comes from a forward-backward diffusion framework: the forward process gradually degenerates the data into random noise with a $T$-step noise schedule (typically, $T = 1000$ as default), while the backward process learns a neural network to iteratively estimate and remove the noise added to the data. However, to generate high quality samples, DMs usually require hundreds of sampling steps (i.e., function evaluations of the trained model). The slow sampling efficiency incurs heavy computational overhead at inference, especially to large-scale DMs such as DALL-E (Ramesh et al., 2022), Imagen (Saharia et al., 2022) and Stable Diffusion (Rombach et al., 2022; Podell et al., 2024), posing a great challenge for the deployment of DMs.

Recently, there have been lots of research efforts aiming to design fast samplers for DMs, which can be grouped into two major categories. The first category focuses on evolving more advanced formulations for the sampling process that enjoy faster convergence. Denoising diffusion implicit models (DDIM) (Mohamed & Lakshminarayanan, 2016; Song et al., 2021a), stochastic differential equations (SDE) (Song et al., 2021b) and ordinary differential equations (ODE) based solvers (Zhang & Chen, 2023; Lu et al., 2022) are representative ones. It is worth noting that the ODE samplers allow to generate high quality samples in tens of sampling steps. The second category relies on knowledge distillation schemes, such as progressive distillation (Salimans & Ho, 2022), two-stage progressive distillation (Meng et al., 2023) and consistency distillation (Song et al., 2023; Luo et al., 2023), by which the few-step samples generated by a student DM using the distilled sampler can match to the many-step outputs of its corresponding teacher DM.

In this work, *we attempt to improve the sampling efficiency of DMs in a more generalized perspective*. Specifically, *we ask*: given a pre-trained DM (with either U-Net or self-attention based backbone), no matter what kind of existing

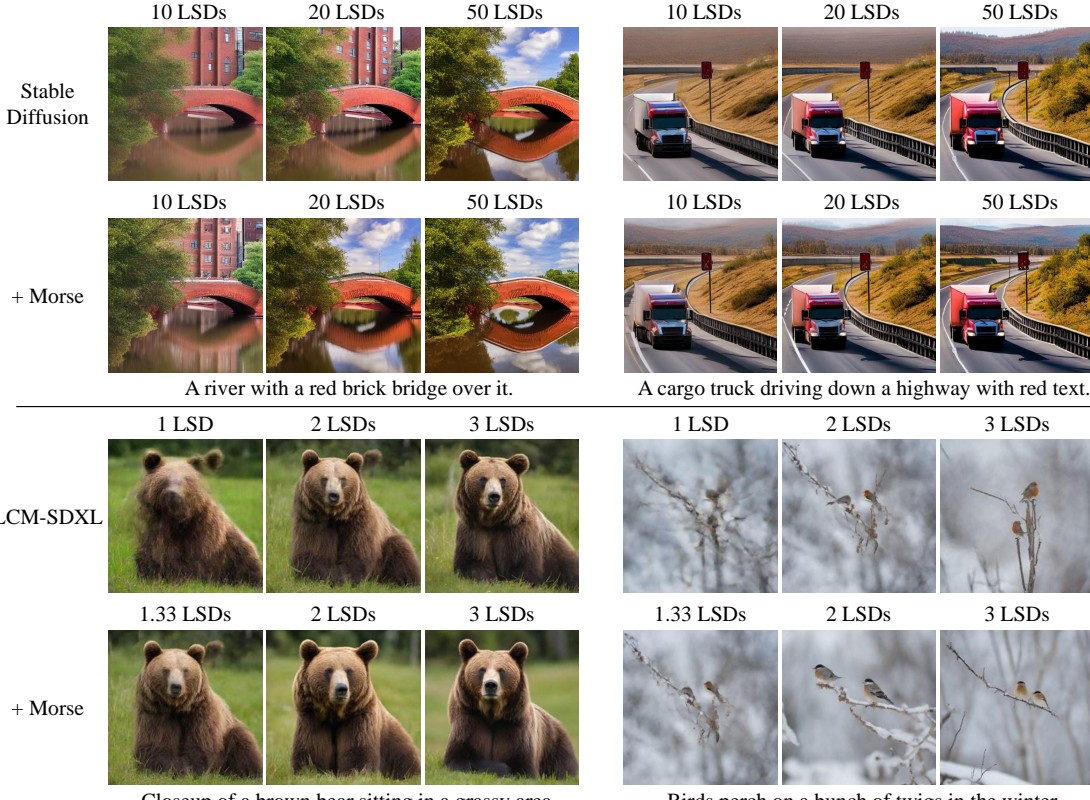

*Figure 1.* Generated samples from Stable Diffusion (Rombach et al., 2022) and Stable Diffusion XL fine-tuned with Latent Consistency Models (LCM-SDXL) (Luo et al., 2023) with and without Morse for text-to-image generation. For simplicity, we use the Latency per Sampling step of the baseline DM (LSD) as the time unit to calculate the total latency of a diffusion process.

samplers is used, is it possible to reformulate the iterative denoising generation (from noise to data) process towards better performance-efficiency tradeoffs under a wide range of sampling step budgets (including hundreds-step, tens-step and few-step sampling)? To address this problem, our method is inspired by a common property of prevailing DMs. We notice that they typically support *jump sampling* (JS) in function evaluation, especially when using the fast samplers discussed above. This observation inspires us to explore the use of JS for formulating our method. Not surprisingly, with JS, prevailing DMs can generate samples in a faster speed, yet inevitably leads to worse sample quality due to the information loss over unvisited steps between every two adjacent JS points on the diffusion trajectory. The performance degradation issue becomes more serious as the JS step length increases. Therefore, the double-edged nature of JS prohibits its use for performance lossless acceleration.

We overcome this barrier by presenting *Morse*, a simple diffusion acceleration framework consisting of two models called *Dash* and *Dot* which tactfully couple JS with a novel residual feedback learning strategy, compensating for the information loss and attaining the desired lossless acceleration in terms of image generation quality. In the formulation of Morse: (1) The Dash model is just the pre-trained diffu-

sion model that needs to be accelerated, but operates in a JS regime, creating sufficient space for sampling efficiency improvement; (2) The Dot model is significantly faster (e.g., multiple times faster in latency) than the Dash model, which is learnt to generate residual feedback conditioned on the observations (including input and output samples, time steps and noise estimate) at the current JS point on the trajectory of the Dash model, lifting the noise estimate to closely match the next-step estimate of the Dash model without JS; (3) Morse chains the outputs of the Dash and Dot models run in a time-interleaved fashion, allowing us to easily choose a proper JS step length to attain performance-efficiency tradeoffs under a wide range of sampling step budgets. Intriguingly, as the Dot model is significantly faster than the Dash model, it enables the Dot model to run several times more sampling steps than the Dash model within the interval of two adjacent JS points while enjoying the same speed. Benefiting from this appealing merit, our method can perform more sampling steps under the same sampling step budget relative to the pre-trained target DMs, establishing a strong base to achieve the desired acceleration goal. Besides the strong ability to accelerate DMs, Morse is also efficient for training and inference, thanks to our proposed weight sharing strategy between the Dash and Dot models. In the strategy, we construct the Dot model by adding

extra light-weight blocks to the pre-trained DM and adopt lightweight Low-Rank Adaptation (LoRA) (Hu et al., 2022) for fast training. During the training of the Dot model, the shared weights from the Dash model remain fixed, while the newly added layers and LoRA modules are trained jointly. On six public image generation benchmarks, our method achieves promising results under lots of experimental setups. In Fig. 1, we show illustrative text-to-image generation results using Stable Diffusion and LCM-SDXL with and without Morse under different sampling-step budgets.

## 2. Method

### 2.1. Background and Motivation

**Basic Concept.** A diffusion model (DM) can generate high quality images. It consists of a forward process for converting image to noise and a generation process (i.e., reverse process) for converting noise to image, both of which are typically formulated as Markov chains with $T$ time steps in total. In the forward process, an image $\mathbf{x}_0 \in \mathbb{R}^{h \times w \times c}$ is first sampled from a data distribution $\mathcal{D}$. At the $t$-th time step, the sample $\mathbf{x}_t$ is added with a random noise $\epsilon \sim \mathcal{N}(0, I)$ having the same dimension, which produces $\mathbf{x}_{t+1}$ for the next time step $t+1$. The distribution for $\mathbf{x}_t$ conditioned on $\mathbf{x}_0$ can be represented as:

$$p(\mathbf{x}_t|\mathbf{x}_0) = \int (p(\mathbf{x}_0) \prod_{i=1}^{t} p(\mathbf{x}_i|\mathbf{x}_{i-1})) d\mathbf{x}_{1:t-1}, \quad (1)$$

where $p(\mathbf{x}_0) \sim \mathcal{D}$ and $p(\mathbf{x}_i|\mathbf{x}_{i-1})$ corresponds to the parameterized function for adding noise. As $t$ increases, $\mathbf{x}_t$ gets noisier, where $\mathbf{x}_T$ conforms to the distribution $\mathcal{N}(0, I)$. With the forward process, a neural network $\theta$ is trained to estimate the original image $\mathbf{x}_0$ (equivalent to estimate noise $\epsilon$) from any time step $t$:

$$\mathbf{z}_t = \theta(\mathbf{x}_t, t), \quad (2)$$

where $\mathbf{z}_t$ denotes the estimate generated by the trained network $\theta$ for approximating $\mathbf{x}_0$. Now, we can use $\theta$ to reverse the forward process from noising to denoising for image generation. Specifically, in the generation process, a noise $\epsilon \sim \mathcal{N}(0, I)$ is firstly sampled as $\mathbf{x}_T$. With the estimate $\mathbf{z}_t$ from $\theta$, we can approximate the distribution of $p(\mathbf{x}_{t-1}|\mathbf{x}_t)$ by $p(\mathbf{x}_{t-1}|\mathbf{x}_t, \mathbf{x}_0 = \mathbf{z}_t)$ using Bayes' rule and Eq. 1:

$$p(\mathbf{x}_{t-1}|\mathbf{x}_t) \approx \frac{p(\mathbf{x}_t|\mathbf{x}_{t-1})p(\mathbf{x}_{t-1}|\mathbf{x}_0 = \mathbf{z}_t)}{p(\mathbf{x}_t|\mathbf{x}_0 = \mathbf{z}_t)}. \quad (3)$$

Therefore, we can iteratively convert a noise $\mathbf{x}_T$ to an image $\mathbf{x}_0$ along the time step from $T$ to 0 with $p(\mathbf{x}_0|\mathbf{x}_T) = \int (p(\mathbf{x}_T) \prod_{i=1}^{T} p(\mathbf{x}_{i-1}|\mathbf{x}_i)) dx_{1:T-1}$. So far, we can generate high quality images with $T$ sampling steps following Eq. 3. However, such a generation process is very time-consuming. At each of $T$ steps, the trained network $\theta$ needs

to evaluate for one time. While the number of total steps $T$ is mostly very large (e.g., $T = 1000$ for DDPM (Ho et al., 2020)). It is essential for the generation process to well approximate the reverse of the forward process (Sohl-Dickstein et al., 2015; Ho et al., 2020; Song et al., 2021a).

**Jump Sampling.** For a better sampling efficiency, most prevailing DMs adopt the jump sampling (JS) strategy, in which not all the time steps $T, \ldots, 0$ but only a decreasing subsequence of them are visited. We denote the sub-sequence as $t_n > \cdots > t_0 (t_i \in [0, T])$, mostly sampled uniformly from $T$ to 0. Therefore, the number of visited steps $n$ can be much smaller than the total number of time steps $T$, leading to a faster speed for the generation process. With JS, each sampling step can be represented as:

$$\mathbf{x}_{t_{i-1}} = \phi(\mathbf{x}_{t_i}, \mathbf{z}_{t_i}, t_i, t_{i-1}), \quad (4)$$

where $\phi$ is the schedule function used to update the sample from $\mathbf{x}_{t_i}$ to $\mathbf{x}_{t_{i-1}}$, which is defined according to different samplers (e.g., DDPM (Ho et al., 2020), DDIM (Song et al., 2021a), SDE (Song et al., 2021b), DPM-Solver (Lu et al., 2022), CM (Song et al., 2023)). Intuitively, for a generation process, the neighboring steps tend to have similar sample $\mathbf{x}_t$ and close time step $t$ as inputs for $\theta$, leading to similar estimate $\mathbf{z}_t$. So that with the estimate $\mathbf{z}_{t_i}$, the sample can jump over multiple steps toward the same estimate from $t_i$ to $t_{i-1}$, without doing much harm to the sample quality. As more steps are jumped over, the step length between two adjacent JS points becomes longer and the performance degradation issue becomes more serious. Therefore, the double-edged nature of JS prohibits its use for performance lossless acceleration, while it also leaves room for us to further improve it. If we can efficiently reduce the information loss caused by JS while maintaining its high sampling efficiency, then we can achieve a better performance-efficiency tradeoff. This is the key motivation of our work.

### 2.2. Morse

As we discussed above, our key motivation is to efficiently reduce the information loss caused by JS while maintaining the high sampling efficiency. To achieve this goal, we present Morse, a simple diffusion acceleration framework with dual-sampling, as illustrated in Fig. 2. With Morse, the generation process is reformulated from iteration with a single model to interaction between two models, which are called *Dash* and *Dot*.

**Formulation of Morse.** The Dash model is just the pre-trained DM, but operates in a JS regime, creating sufficient space for sampling efficiency improvement. The Dot model is newly introduced by us for accelerating the Dash model, which is $N$ times faster than the Dash model. During the generation process, each sampling step is either with noise estimate from the Dash model or the Dot model, while the

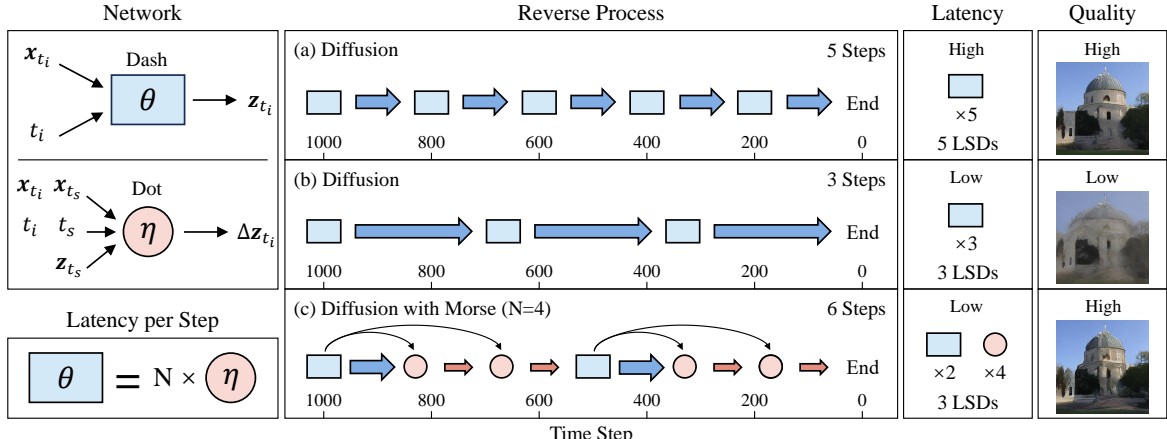

*Figure 2.* Illustration of diffusion with Morse. Morse consists of two models named Dash and Dot, which interact with each other during the generation process. Dash is the pre-trained model of any type to be accelerated, which operates in a jump sampling regime. Dot is the model newly introduced by us to accelerate Dash, which is $N$ times faster than Dash in latency. We provide examples to show how our Morse works. For simplicity, we use the Latency per Step of the baseline DM (LSD) as the time unit to calculate the total latency of a diffusion process. (i.e., the latency per step of the Dot model is mapped to that of the baseline Dash model) (a) Standard generation process, which performs 5 steps under 5 LSDs; (b) Standard generation process, which performs 3 steps under 3 LSDs; (c) Generation process with Morse, which performs 6 steps under 3 LSDs. Under the same latency, a generation process with Morse can perform more steps and achieve better sample quality.

two models play different roles. As we described in Sec. 2.1, the Dash model can estimate noise independently. The Dot model learns to generate residual feedback conditioned on the observations (including input and output samples, time steps, and noise estimate) at the current sampling point on the trajectory of Dash, lifting the noise estimate to closely match the next-step estimate of the Dash model without JS. Morse chains the outputs of the Dash and Dot models run in a time-interleaved fashion. For a generation process with Morse, we reformulate how to estimate noise as:

$$
\mathbf{z}_{t_i} = \begin{cases} \theta(\mathbf{x}_{t_i}, t_i) & t_i \in S \\ \mathbf{z}_{t_s} + \eta(\mathbf{x}_{t_s}, \mathbf{x}_{t_i}, \mathbf{z}_{t_s}, t_s, t_i) & t_i \notin S \end{cases}, \quad (5)
$$

where $\theta$ denotes the Dash model; $\eta$ denotes the Dot model; $t_s$ denotes the current sampling point on the trajectory of the Dash model when the Dot model produces noise estimation at the step $t_i$; $S = \{t_{s_d}, \ldots, t_{s_1}\}$ denotes the set of sampling steps with the noise estimates from the Dash model, which is a sub-sequence of $t_n, \ldots, t_0$. The above formulation of Morse is simple and easy to implement, and has the great capability to accelerate diffusion models generally as tested with various experimental settings.

**Weight Sharing between Dash and Dot.** To reduce the training and computational costs of the Dot model, we introduce a weight sharing strategy between Dash and Dot. As shown in Fig. 3, we construct the Dot model by adding $m$ trainable lightweight down-sampling blocks and up-sampling blocks on the top and under the bottom of the pre-trained Dash model respectively. The extra blocks have the significantly reduced numbers of channels and layers compared with the pre-trained blocks. For each of the pre-

trained blocks, the resolution of its input is reduced by $4^m$ times. Therefore, the Dot model can be significantly faster than the Dash model. When training the Dot model, we fix the shared pre-trained layers and adopt lightweight Low-Rank Adaptation (LoRA) (Hu et al., 2022) for quickly adapting to the new training objective and resolutions. With this simple and low-cost design, our Dot model can be derived from the pre-trained DM very efficiently, since it reserves nearly all the knowledge learned by the Dash model.

### 2.3. A Deep Understanding of Morse

To have a deep understanding of how Morse can improve the sampling efficiency of DMs, we give detailed explanations in two perspectives.

**How can Morse Accelerate Different Diffusion Models?** With JS, DMs can generate samples in a faster speed, yet inevitably lead to worse sample quality due to the information loss over unvisited steps between two adjacent JS points on the diffusion trajectory. To compensate for the information loss, we insert extra multiple sampling points with Dot between every two adjacent JS points, which efficiently reduces the JS step length. Since Dot is $N$ times faster than Dash, the inserted sampling steps can be completed by Dot with only $1/N$ time budget compared with Dash. In other words, Morse can perform more sampling steps under the same sampling step budget relative to the pre-trained DMs. We assume a standard generation process has $n$ sampling steps. Under the same latency (for $n$ sampling steps of baseline DMs), there could be $n - k$ ($0 \le k < n$) sampling steps with Dash and $Nk$ sampling steps with Dot in our Morse, which introduces $(N - 1)k$ extra sampling steps. Given a

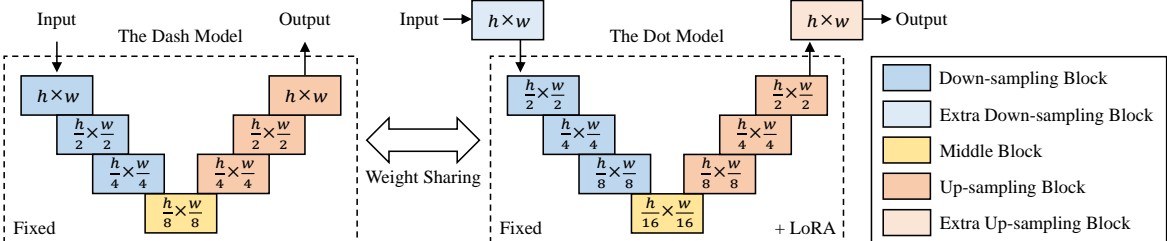

*Figure 3.* Illustration of weight sharing between Dash and Dot. The Dot model is constructed by adding $m$ ($m = 1$ for the illustrated example) trainable lightweight down-sampling and up-sampling blocks on the top and under the bottom of the pre-trained Dash model respectively. $h \times w$ denotes the resolution of input feature maps. When training the Dot model, we fix the shared pre-trained layers and add lightweight Low-Rank Adaptation (LoRA) to help the Dot model for fast convergence.

specific sampling step budget, Morse can flexibly change the JS step length by controlling $k$. Under ideal conditions where Dot and Dash perform exactly the same for noise estimation, this leads to a speedup of $(n - k + Nk)/n$, which is the upper bound speedup for our Morse.

**How can Dot Behave as Dash on Noise Estimation?** To answer the question, we reiterate our design of the Dot model: (1) Dot cooperates with Dash by learning to generate residual feedback utilizing the trajectory information; (2) Dot inherits most of pre-trained weights from Dash. When training the Dot model, we fix the shared pre-trained layers and add LoRA to help the Dot model for fast convergence. Benefiting from the first design, Dot does not need to estimate noise independently but generates residual feedback conditioned on the observations at the current sampling point on the trajectory of Dash. By using the input sample, output sample, time steps and noise estimate as inputs, Dot gets the information about how the sample is updated between the two sampling steps, which largely helps Dot on adjusting the noise estimate of Dash. In the second design, we adopt a weight sharing mechanism between Dash and Dot. It allows Dot to inherit most of the knowledge learned by Dash, which guarantees the consistency between Dash and Dot in the residual learning process. Additionally, the weight sharing mechanism also improves the parameter efficiency and training efficiency of Morse. By adding extra lightweight trainable blocks to a pre-trained DM, the Dot model can be trained very efficiently with LoRA. Thanks to the adaptive residual feedback strategy with trajectory information and weight sharing mechanism, Dot is able to easily lift the noise estimate at the current JS point to closely match the next-step estimate of the Dash model. Since JS strategy is adopted by most popular DMs, our Morse can widely accelerate various DMs with different samplers, benchmarks, and network architectures under diverse sampling step budgets, as we show in what follows.

**Difference with the distillation-based methods.** From the perspective of learning the knowledge from a pre-trained model, Morse is somehow similar with the distillation-based methods for diffusion. While they are different both in formulation and focus: (1) With Morse, the generation process

is reformulated as interaction between the Dash and Dot models, rather than iteration with a student DM; (2) Morse adopts an adaptive residual feedback strategy with trajectory information; (3) The aim of Morse is to efficiently reduce the information loss caused by jump sampling to attain the lossless acceleration goal. In distillation-based methods, a student DM is trained to match the outputs of its corresponding teacher DM in a sampling process using much fewer steps, but always with performance degradation issue; (4) Morse is complementary to the distillation-based methods, which can be used to further accelerate a DM trained with distillation, as we show in the experiments.

## 3. Experiments

### 3.1. Metric to Evaluate Speedup

**Speedup.** Before showing the experimental results, we first describe how we evaluate the speedup of Morse. For a pre-trained DM, we assume two generation processes with and without Morse. The total latency of the process without Morse is $n$ and the total latency of the process with Morse is $l(n \geq l)$. The two processes get the same evaluation metric. Then, the speedup of Morse under the latency of $l$ can be calculated as $n/l\times$.

For a diffusion model (DM), we first measure its sample quality with and without Morse under different latencies, mainly using the mostly adopted metric Fréchet inception distance (FID, lower is better) (Heusel et al., 2017). The sampling steps are selected following the official settings. Then, we use linear interpolation to fit the curves between latency and evaluation metrics for approximating the evaluation metric under any available latency. Note that it's too time-consuming to evaluate the metrics with all the latencies. To be intuitive, we calculate an average speedup of Morse over the selected latencies. We fit a curve between a set of latencies and speedups to approximate speedups across all the latencies. All the speeds for different models are tested using an NVIDIA GeForce RTX 3090. Recall that Dot is $N$ times faster than Dash. The speeds of the models may vary on different GPUs, leading to the change of $N$ and speedup. While we find that a Dash model and

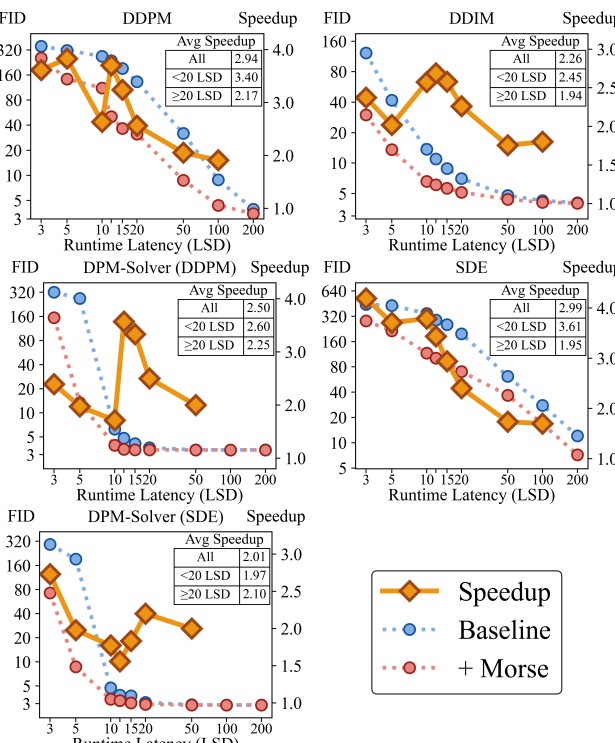

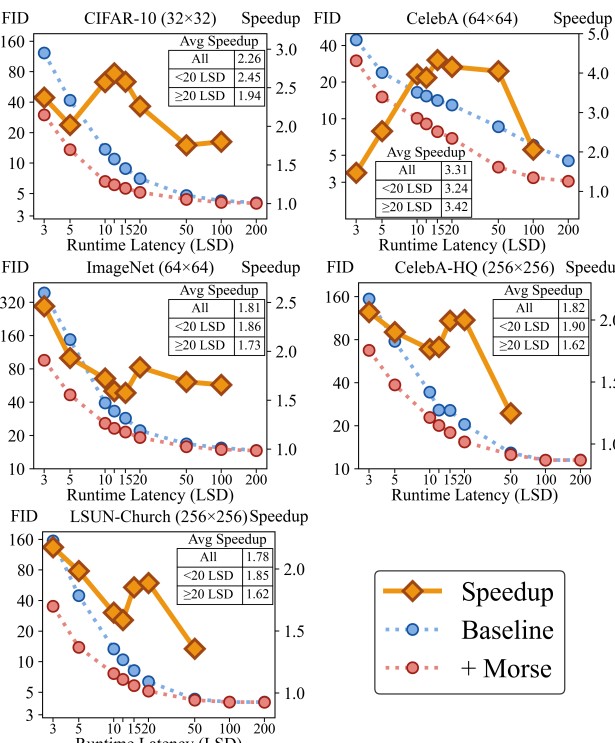

*Figure 4.* Results of Morse with different samplers on CIFAR-10 benchmark. A speedup of $n$ under the latency of $l$ means that the DM with Morse under $l$ and the DM without Morse under $nl$ achieve the same FID. We calculate the average speedups over all the latencies, latencies from 3 to 15 LSDs and 20 to 100 LSDs.

*Figure 5.* Results of Morse with DDIM sampler on different image generation benchmarks.

its Dot model mostly have little change in $N$ on different GPUs, our Morse demonstrates a good acceleration ability consistently. *Details are provided in the Appendix.*

**LSD.** For simplicity and generalization, we normalize the total latency of a diffusion process with the time unit *Latency per Step of the baseline DM (LSD)*, namely the time cost of the Dash model for one sampling step (i.e., the latency per step of the Dot model is mapped to that of the baseline Dash model). It takes 1 LSD for Dash and $1/N$ LSD for Dot to perform one step. For a diffusion process without Morse under $n$ sampling steps, its latency (namely the end-to-end time for generation images) can be represented as $n$ LSDs.

### 3.2. Accelerate Image Generation

**Experimental Setup.** For each DM evaluated in experiments, we collect its official pre-trained model as the Dash model, of which the weights are fixed. With the weight sharing strategy, all the Dot models are trained following the official training settings, while typically with reduced batch size and training iterations. We typically set the number of extra down-sampling blocks and up-sampling blocks $m$ to 2, leading to $N$ in the range of 5 to 10. All the experiments are performed on the servers having 8 NVIDIA GeForce RTX 3090 GPUs. *More details are described in the Appendix.*

**Different Samplers.** In the experiments, we evaluate our Morse with the mainstream samplers, including DDPM (Ho et al., 2020), DDIM (Song et al., 2021a), DPM-Solver (Lu et al., 2022) for discrete samplers and SDE (Song et al., 2021b), DPM-Solver on SDE for continuous samplers. We conduct the experiments with CIFAR-10 (Krizhevsky, 2009) benchmark, which is adopted by all the above samplers for experiments. As shown in Fig. 4, our Morse can accelerate DMs consistently with all the samplers under different LSDs ranging from 3 to 100, achieving average speedups ranging from $2.01\times$ to $2.94\times$. The results also show that our Morse can work with both discrete-time and continuous-time methods. Morse can even significantly accelerate the state-of-the-art sampler DPM-Solver, which can generate high quality images with very few steps by also utilizing the trajectory information from previous steps. Note that we calculate the speedups of Morse as $N/A$ for DPM-Solver on both DDPM and SDE with 100 LSDs, which are not used for calculating the average speedups. The reason is that there is no room to accelerate, since the FIDs constantly keep the same (even worse) value with latencies larger than 100 LSDs for the baseline DMs. *The results with different samplers on other benchmarks are provided in the Appendix.*

**Different Benchmarks.** In the experiments, we further evaluate our Morse with 5 popular image generation benchmarks, including CIFAR-10 (32×32) (Krizhevsky, 2009), ImageNet (64×64) (Russakovsky et al., 2015), CelebA (64×64) (Liu et al., 2015), CelebA-HQ (256×256) and

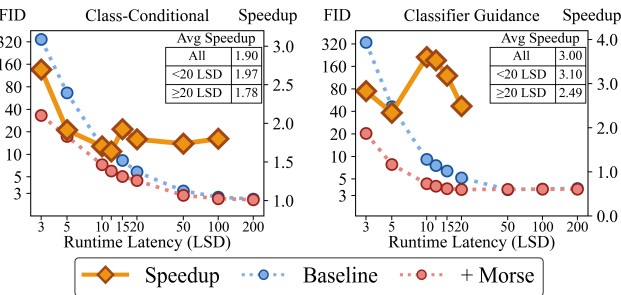

*Figure 6.* Results of Morse with different conditional generation strategies on ImageNet benchmark.

LSUN-Church (256×256) (Yu et al., 2015). Since we have evaluated Morse with different samplers, we keep the sampler as the most widely used DDIM in the following experiments unless otherwise stated, to exclude the impact of differences in samplers. The results are shown in Fig. 5. Our Morse can be generalized well to all the benchmarks, which have different image resolutions (from 256×256 for LSUN-Church and CelebA-HQ to 32×32 for CIFAR-10), different dataset sizes (from 1.2 million for ImageNet to 30 thousand for CelebA-HQ) and different semantic information. For all the benchmarks under most LSDs, our Morse gets speedups around 2×. On the CelebA, it can even achieve speedups more than 4× under some LSDs.

**Different Conditional Generation Strategies.** After showing the effectiveness of Morse under unconditional generation, we next evaluate our Morse under conditional generation with different strategies, including class-conditional and classifier-guided image generation (Ho & Salimans, 2021) on ImageNet benchmark at resolution 64×64. For the classifier guidance, we consider the classifier as a part of Dash and train the Dot to approximate the estimate guided by a classifier. As shown in Fig. 6, Morse can well generalize to conditional generation with different strategies.

**Different Network Architectures.** In the above experiments, there are 8 different network architectures collected from 6 research works with model sizes ranging from 35.75M to 421.53M for the Dash models (Rombach et al., 2022; Ho et al., 2020; Nichol & Dhariwal, 2021; Song et al., 2021a;b; Dhariwal & Nichol, 2021). It can be seen that our Morse achieves good generalization ability under all diffusion architectures with different capacities and model sizes.

### 3.3. Accelerate Text-to-Image Generation

Next, we evaluate our Morse under the highly popular text-to-image generation task with the latent-space Stable Diffusion model (Rombach et al., 2022).

**Experimental Setup.** We select the Stable Diffusion v1.4 as our Dash model, which is pre-trained with around 2 billion text-image pairs from LAION-5B dataset (Schuhmann et al., 2022). In our experiments, the Dot model is trained

*Table 1.* FIDs of Stable Diffusion with and without Morse on MS-COCO. We calculate FIDs under different classifier-free guidance scales and select the best FID among all the scales and FID under default scale of 7.5 for comparison.

| Method | FID | Latency (LSD) | | | |
|---|---|---|---|---|---|
| | | 10 | 15 | 20 | 50 |
| Stable Diffusion | scale of 7.5 | **11.75** | 11.92 | 12.35 | 13.53 |
| | best scale | 10.65 | 9.47 | 8.70 | **8.22** |
| + Morse | scale of 7.5 | **9.29** | 10.07 | 10.93 | 13.22 |
| | best scale | 8.60 | 8.55 | 8.29 | **8.15** |

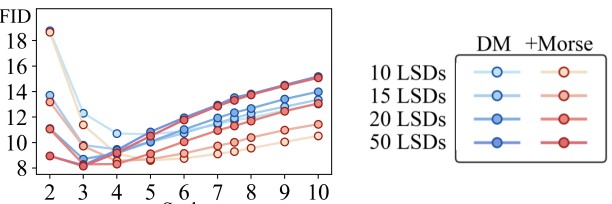

*Figure 7.* Stable Diffusion with and without Morse under different latencies and scales.

with only about 2M text-image pairs at resolution 512×512 sampled from the LAION-5B dataset. We use DDIM as the sampler. Following the popular evaluation protocol, we adopt the FID (lower is better) and CLIP score (Radford et al., 2021) (higher is better) as the evaluation metrics and use the 30000 generated samples with the prompts from the MS-COCO (Lin et al., 2014) validation set for evaluation. The CLIP scores are calculated using ViT-g/14. All the experiments are performed on a server having 8 NVIDIA Tesla V100 GPUs. *More details are described in the Appendix.*

**Results Comparison.** Following the default settings, we first evaluate the FIDs of Stable Diffusion with and without Morse, using the classifier-free guidance scale of 7.5. While we find that increasing the number of steps does not always lead to consistently better FID scores for standard Stable Diffusion, as shown in Table 1. Therefore, we find another two schemes to evaluate speedups. In the first scheme, we evaluate the FID with different scales and select the best FID score for comparison. From the results shown in Fig. 7, we can find that the best FID consistently gets better when the latency increases. Under this scheme, we can calculate an average speedup of 2.29×. In the other evaluation scheme, we fit the curves between FID and CLIP scores under different scales using the linear interpolation following Stable Diffusion. The results are shown in Fig. 8. When the scale is larger than 4, we can observe a tradeoff between the two metrics. The curve with our Morse is below the curve without Morse. For example, the curve with Morse under 10 LSDs is below the curve without Morse under 20 LSDs in most scales, indicating an average speedup of approximately 2×. Some generated samples for comparison are provided in Fig. 1 and Appendix. The results further demonstrate the generalization ability of Morse. On the popular text-to-image generation task with a large DM (859.52M), our Morse still shows a significant acceleration ability. As

*Table 2.* Training details of Stable Diffusion and the corresponding Dot model in Morse. The training memory cost is tested with the batch size of 8 per GPU.

| Model | Params (M) | Training Samples (M) | Training Cost (A100 hours) | Training Memory (MB) |
|---|---|---|---|---|
| Stable Diffusion | 859.52 | 2,000 | 150,000 | 23,485 |
| Dot model | 97.84 (+11.4%) | 2 (+0.1%) | 190 (+0.1%) | 18,841 (-19.8%) |

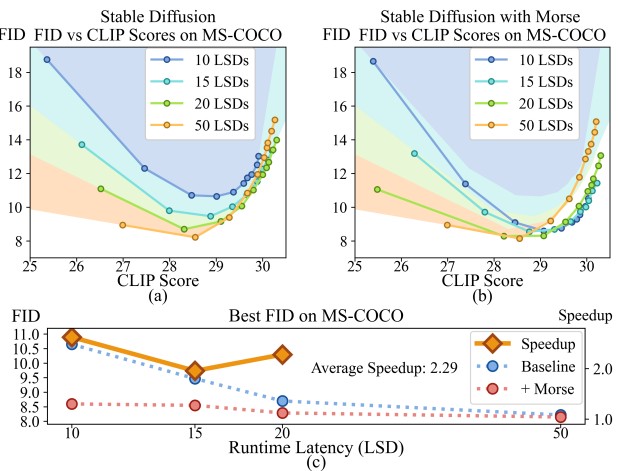

*Figure 8.* Results of Morse with Stable Diffusion. (a) and (b) are curves between FIDs and CLIP scores for Stable Diffusion with and without Morse on different LSDs under guidance scales of 2, 3, 4, 5, 6, 7, 7.5, 8, 9, 10, which correspond to the points in the curves from left to right. We paint the background using the curve of standard Stable Diffusion for better illustration; (c) Curves between FIDs and LSDs using the best FIDs among different scales.

shown in Table 2, while the Dash model is trained with heavy computational resources and large datasets, our Dot can be trained very efficiently with less than 0.1% text-image pairs and 0.1% training cost compared with it. With the trajectory information, the Dot model can be easily close to the Dash model on noise estimation. The results also show that our Morse works well with the classifier-free guidance and the latent-space diffusion models.

### 3.4. Ablation Study and More Comparisons

**Effect of Trajectory Information.** Recall that our core insight is that the trajectory information can help the Dot model to perform as well as the Dash model without JS on noise estimation. In Morse, we use the sample $\mathbf{x}_{t_s}$, the time step $t_s$ and the noise estimate $\mathbf{z}_{t_s}$ at the current sampling point on the trajectory of Dash as the extra inputs for Dot. In the experiments, we evaluate Morse with different combinations of them with DDIM sampler on CIFAR-10 benchmark under 10 LSDs. From the results shown in Table 3, we can see that each of the inputs is helpful for Dot on residual estimation. Without the trajectory information, the introduction of the Dot model can not accelerate DMs anymore, because of its inferior estimation. These ablative results prove that the trajectory information plays a key role in our design, which also validate our key insight to some extent.

*Table 3.* Ablation of Morse with different trajectory information.

| Method | $\mathbf{x}_{t_s}$ | $\mathbf{z}_{t_s}$ | $t_s$ | FID |
|---|---|---|---|---|
| DDIM | - | - | - | 13.67 |
| + Morse | | | | 13.56 |
| | | | ✓ | 8.11 |
| | | ✓ | | 8.06 |
| | | ✓ | ✓ | 7.60 |
| | ✓ | | | 7.50 |
| | ✓ | | ✓ | 6.83 |
| | ✓ | ✓ | | 7.27 |
| | ✓ | ✓ | ✓ | **6.60** |

*Table 4.* CLIP scores of LCM-SDXL with and without Morse on MS-COCO.

| Method | LSD | CLIP score |
|---|---|---|
| LCM-SDXL | 1 | 25.39 |
| | 2 | 29.40 |
| | 3 | 30.34 |
| | 4 | **30.80** |
| LCM-SDXL with Morse | 1.33 | 28.70 |
| | 1.67 | 29.84 |
| | 2 | 30.30 |
| | 3 | **30.83** |

**LCM-SDXL with Morse.** In the above experiments, we have demonstrated the great capability of Morse to accelerate 9 baseline diffusion models on 6 image generation tasks. Here, we further show that Morse can be also generalized to improve Latent Consistency Models (LCM) (Luo et al., 2023), which is a popular distillation-based method tailored for few-step text-to-image synthesis. We use LCM-SDXL as the baseline, which denotes a Stable Diffusion XL model fine-tuned with LCM. The resolution is 1024×1024. We evaluate LCM-SDXL with Morse on MS-COCO benchmark as described in Sec 3.3. Same with Stable Diffusion, Stable Diffusion XL also adopts the classifier-free guidance, while LCM fixes the scale to 7.5 during the distillation. Under the fixed scale of 7.5, for standard LCM-SDXL, we notice that its FID score does not consistently get better as the number of steps increases, while the CLIP score does. Therefore, we select the CLIP score as the metric for evaluating LCM-SDXL with Morse. The results are shown in Table 4. Since the Dot model is 3 times faster than the Dash model, we can evaluate the CLIP scores for LCM-SDXL with Morse under LSDs of 1.33 and 1.67. Over a sampling step number from 1 to 4, we can calculate an average speedup of 1.43× on the CLIP score for our Morse. *Experimental details and some generated samples for comparison are provided in the Appendix.*

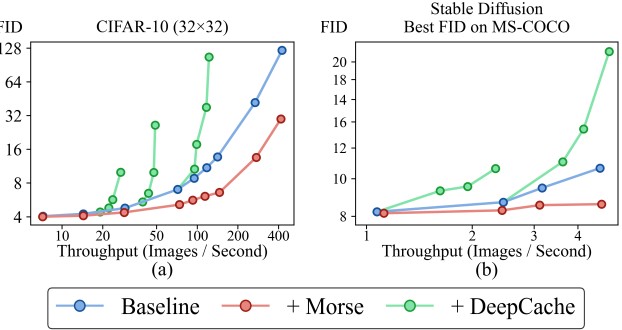

*Figure 9.* Comparison between Morse and DeepCache with Stable Diffusion v1.4 using DDIM sampler. To give a clear comparison, we report the FIDs of two methods under different throughputs, which are evaluated on an NVIDIA GeForce RTX 4090 GPU with the batch size of 20. The results are tested with the public code of DeepCache. Following the official settings of DeepCache, we evaluate it with the number of steps 20, 50, 100 under $N = 2, 3, 5, 10$ on CIFAR-10 and the number of steps 20, 50 under $N = 2, 3, 5$ on MS-COCO.

*Table 5.* FIDs of Morse and PFDiff on MS-COCO benchmark with Stable Diffusion. The results of PFDiff are collected from the paper. Following the default settings of PFDiff, we evaluate Morse with 10000 generated samples under the guidance scale of 7.5.

| Method | Latency (LSD) | | | |
|---|---|---|---|---|
| | 6 | 10 | 15 | 20 |
| Stable Diffusion | 20.33 | 16.78 | 16.08 | 15.95 |
| + PFDiff | 15.47 | 13.06 | 13.57 | 13.97 |
| + Morse | **13.88** | **11.67** | **12.22** | **13.63** |

*Table 6.* FIDs of Morse and AYS on ImageNet ($64 \times 64$) benchmark. For fair comparisons, we collect the results of AYS from the paper and evaluate Morse under the same settings.

| Method | Latency (LSD) | | |
|---|---|---|---|
| | 5 | 10 | 15 |
| DDIM | 147.44 | 39.40 | 28.68 |
| + AYS | 50.38 | 29.23 | 24.21 |
| + Morse | **46.63** | **25.76** | **21.50** |

**Comparison with Feature Reuse.** To further demonstrate the effectiveness of Morse, we compare Morse with the state-of-the-art feature reuse methods, including DeepCache (Ma et al., 2024) and PFDiff (Wang et al., 2025). Same to Morse, the methods also explore the temporal step redundancies for accelerating diffusion models, while in different ways. Specifically, DeepCache reuses the features at step $t$ for $N - 1$ following steps, and PFDiff utilizes the states of past steps stored in a buffer to update the states of future steps. As shown in Figure 9, we can find that Morse consistently gets better throughput and FID than DeepCache on both benchmarks. For acceleration, DeepCache is lossy in generation quality due to reusing most of the features at step for $N - 1$ following denoising steps, yet Morse is lossless. From the results shown in Table 5, we can find that Morse is also superior to PFDiff.

**Comparison with Time Step Schedule Optimization.** To accelerate diffusion models, time step schedule optimization methods design different strategies to choose optimal time steps when given a small number of sampling steps. The line of work is also closely related to Morse. In Table 6, we provide the comparison results between Morse and AYS (Sabour et al., 2024), which is a state-of-the-art method in this domain. As shown, Morse performs better than AYS over different sampling step budgets.

**More Ablations and Discussions.** In the Appendix, we provide more ablative experiments and discussions about Morse for a better understanding, including: (1) The runtime latencies of Dash models and Dot models on different GPUs; (2) The principles to determine the scheduling of Morse; (3) The performance of Morse under a specific latency using different numbers of steps with Dash; (4) Comparison between different architectural designs for the Dot model; (5) Other results and more generated samples.

# 4. Related Work

Besides the fast samplers discussed in the Introduction section, there are other emerging efforts to speed up the inference of DMs. Some recent works use quantization (Li et al., 2023b; Chen et al., 2023b), pruning (Li et al., 2022; Wang et al., 2024), reuse of parameters and feature maps (Agarwal et al., 2024; Wimbauer et al., 2024; Ma et al., 2024), time step schedule optimization (Watson et al., 2021; Xue et al., 2024; Sabour et al., 2024), and GPU-specialized optimization (Chen et al., 2023c; Li et al., 2024) to reduce runtime model latency. Another line of research (Li et al., 2023c; Xu et al., 2024; Li et al., 2023a) seeks to design lightweight network architectures for DMs, enabling to deploy them on mobile devices. In design, our method is orthogonal to these methods, and thus it should be able to combine with them for improved performance.

The idea of using dual-model designs to strike a better accuracy-efficiency tradeoff is popular in both computer vision and natural language processing. SlowFast network (Feichtenhofer et al., 2019), a powerful and efficient architecture for video action recognition, uses a slow pathway operating at a low frame rate with low resolution to encode spatial semantics, and a parallel fast pathway operating at a higher frame rate with higher resolution to encode motion cues. Speculative decoding (Stern et al., 2018), a fast decoding mechanism for accelerating the inference of autoregressive language models, predicts candidate tokens with a small approximation model, and verifies the acceptability of these candidate tokens by a larger and powerful target model with a single forward pass, significantly reducing the computation for accepted tokens. Many variants (Li et al., 2020; Leviathan et al., 2023; Chen et al., 2023a; Zhang et al., 2024) of them have been proposed. Although our method is also a dual-model design, it focuses on accelerating diffusion models with a simple framework, and its key insight is to reformulate the iterative generation (from noise to data) process via taking advantage of fast jump sampling and adaptive residual feedback strategies. Clearly, our method differs from them in focus, motivation and formulation.

# 5. Discussion and Conclusion

We present a simple dual-sampling framework called Morse to accelerate diffusion models losslessly. Morse reformulates the iterative generation process by involving two models called Dash and Dot that interact with each other, which exhibits the merit of flexibly attaining high-fidelity image generation while improving overall sampling efficiency. Experimental results show that Morse can generally accelerate diffusion models under various settings. While Morse shows the general acceleration ability, it introduces an extra Dot model with a small number of trainable parameters.

## Impact Statement

As an acceleration method for diffusion models, Morse has broader impacts similar to most generative AI models. For example, it may be misused to help creating realistic fake news and videos to spread false information.

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

# A. Appendix

## A.1. Benchmarks and Evaluation Details

**Image Generation.** In the experiments described in Sec. 3.2, we consider 5 mainstream image generation benchmarks with various resolutions for evaluating the generalization ability of our Morse, including CIFAR-10 (32×32, 50 thousand images) (Krizhevsky, 2009), CelebA (64×64, 0.2 million images) (Liu et al., 2015), ImageNet (64×64, 1.2 million images) (Russakovsky et al., 2015), CelebA-HQ (256×256, 30 thousand images) (Liu et al., 2015), LSUN-Church (256×256, 0.1 million images) (Yu et al., 2015). Following the popular evaluation protocol, for each DM, we generate 50000 samples and calculate the FID score between the generated images and the images of the corresponding benchmark. For fair comparisons, we adopt the settings including data processing pipeline and hyperparameters following the corresponding DMs.

**Text-to-Image Generation.** In the experiments for Stable Diffusion v1.4 and LCM-SDXL, we use 2 million text-image pairs sampled from the LAION-5B (Schuhmann et al., 2022) dataset. Following the popular evaluation protocol, we evaluate the text-to-image diffusion models under zero-shot text-to-image generation on the MS-COCO 2014 validation set (Lin et al., 2014) (256×256). All the generated images are down-sampled from 512×512 to 256×256 for evaluation. For each DM, we generate 30000 samples with the prompts from the validation set. The CLIP scores are calculated using ViT-g/14.

## A.2. Implementation Details for Stable Diffusion

**Implementation Details.** For text-to-image generation, we evaluate our Morse with Stable Diffusion v1.4 (Rombach et al., 2022). In the experiments, we use the Dot model with the extra parameters of 97.84M to accelerate the Dash model with the size of 859.52M. The latencies of the Dash model and the Dot model are 0.709 second and 0.082 second per batch respectively ($N = 8.6$), which is tested using a single NVIDIA GeForce RTX 3090 under a batch size of 20. With the official settings, Stable Diffusion v1.4 is pre-trained with around 2 billion text-image pairs at resolution 256×256 and fine-tuned with around 600M text-image pairs at resolution 512×512 from LAION-5B dataset (Schuhmann et al., 2022). We add two trainable down-sampling blocks and up-sampling blocks, with the numbers of channels 96 and 160, on the top and under the bottom of the pre-trained Stable Diffusion to construct the Dot model respectively. We set the rank of LoRA to 64. In our experiments, the Dot model is trained with only about 2M text-image pairs at resolution 512×512 sampled from the LAION-5B dataset for 100,000 iterations. We use DDIM as the sampler.

For conditional image generation, Stable Diffusion v1.4 adopts the classifier-free guidance, which has a parameter called guidance scale to control the influence of the text prompts on the generation process. To ensure that our Morse can also work well with different guidance scales besides the different numbers of steps, we also randomly sample the guidance scales between 2 and 10 during the training procedure. The Dot models are trained on a server with 8 NVIDIA Tesla V100 GPUs. Considering the risk for misuse of the generative models, we use the safety checker module which is adopted by Stable Diffusion project for the released models.

**Latency of Each Block.** It may be not intuitive that we can construct a Dot model with faster speed by adding extra blocks to a DM. Here, we provide the latency of each block for the Stable Diffusion with and without extra blocks in Fig. 10. The state-of-the-art DMs mostly adopt the U-Net architecture with self-attention layers. With the extra blocks on the top and under the bottom of the pre-trained Stable Diffusion, the resolution of the input for each pre-trained block is reduced by 16 times, which significantly reduces the latencies of the pre-trained blocks. Additionally, the extra blocks have the same architecture with the pre-trained blocks while removing the self-attention layers. Since the computational complexity of a self-attention layer grows quadratically with the number of tokens, the pre-trained blocks with high-resolution feature maps have relatively slow inference speeds. By removing the self-attention layers and reducing the number of channels, the latencies of the extra blocks with the high-resolution feature maps are still relatively low. Therefore, the Dot model can be significantly faster than the Dash model.

## A.3. Implementation Details for LCM-SDXL

In the main experiments, we also evaluate our Morse on the Latent Consistency Models (Luo et al., 2023) (LCM-SDXL with 1024×1024 resolution, which is already accelerated with consistency distillation technique). LCM-SDXL can be used for high quality text-to-image generation with very few steps, which is trained with heavy computational resources and large dataset. We add a trainable down-sampling block and up-sampling block on the top and under the bottom of the pre-trained LCM-SDXL respectively. For each of the original pre-trained blocks, the resolution of its input is reduced by

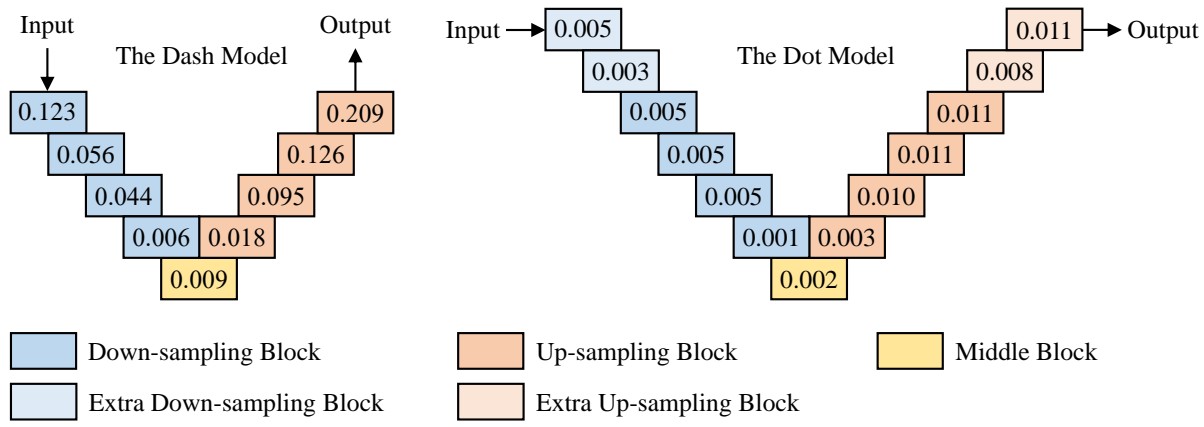

*Figure 10.* Latency (second) of each block for Stable Diffusion with and without adding extra down-sampling and up-sampling blocks. The speeds are tested with the batch size of 20 on a single NVIDIA RTX 3090 GPU.

4 times. The latencies of the Dash model and the Dot model are 0.646 second and 0.211 second per batch respectively ($N = 3.1$), which is tested using single NVIDIA Tesla V100 under a batch size of 5. We fix the shared pre-trained layers except some mismatched layers and add lightweight Low-Rank Adaptation (LoRA) (Hu et al., 2022) to help the Dot model for fast convergence. Compared to the LCM-SDXL with the model size of 2567.55M, the Dot model only has 229.19M trainable parameters, which can be efficiently injected to the Dash model. In our experiments, the Dot model is trained with about 2M text-image pairs at resolution 1024×1024 from the LAION-5B dataset for 100,000 iterations. The Dot model is trained on the servers with 8 NVIDIA Tesla V100 GPUs.

## A.4. Implementation Details for Image Generation

In this section, we provide the implementation details for all the DMs adopted in our experiments for image generation.

*Table 7.* Latency (second) per sampling step of the Dash models and the Dot models on different GPUs. $N$ denotes that the Dot model is $N$ times faster than the Dash model. $h \times w$ denotes the resolution of input feature maps.

| Model Source | Benchmark | RTX 3090 | | | RTX 4090 | | | Tesla V100 | | |
|---|---|---|---|---|---|---|---|---|---|---|
| | | Dash | Dot | N | Dash | Dot | N | Dash | Dot | N |
| DDPM | CIFAR-10 (32×32) | 0.072 | 0.012 | 6.0 | 0.035 | 0.006 | 5.8 | 0.082 | 0.015 | 5.5 |
| | CelebA-HQ (256×256) | 0.539 | 0.112 | 4.8 | 0.346 | 0.073 | 4.7 | 0.680 | 0.135 | 5.0 |
| DDIM | CelebA (64×64) | 0.244 | 0.042 | 5.8 | 0.143 | 0.021 | 6.8 | 0.292 | 0.054 | 5.4 |
| Improved DDPM | ImageNet (64×64) | 0.367 | 0.065 | 5.6 | 0.226 | 0.034 | 6.6 | 0.458 | 0.092 | 5.0 |
| SDE | CIFAR-10 (32×32) | 0.120 | 0.020 | 6.0 | 0.113 | 0.018 | 6.3 | 0.139 | 0.025 | 5.6 |
| LDM | LSUN-Church (256×256) | 0.288 | 0.060 | 4.8 | 0.185 | 0.022 | 8.4 | 0.360 | 0.057 | 6.3 |
| | MS-COCO (512×512) | 0.709 | 0.082 | 8.6 | 0.344 | 0.042 | 8.2 | 0.771 | 0.088 | 8.8 |
| ADM | ImageNet (64×64) | 0.956 | 0.149 | 6.5 | 0.760 | 0.085 | 8.9 | 1.105 | 0.186 | 5.9 |
| ADM-G | | 1.547 | 0.149 | 10.5 | 0.956 | 0.085 | 11.3 | 1.889 | 0.186 | 10.2 |

**Training and Sampling.** In the main experiments, we adopt multiple DMs to evaluate the effectiveness of our Morse, including the models from DDPM (Ho et al., 2020), DDIM (Song et al., 2021a), Improved DDPM (Nichol & Dhariwal, 2021), SDE (Song et al., 2021b), LDM (Rombach et al., 2022) and ADM (Dhariwal & Nichol, 2021). For a DM, we collect its official pre-trained model as the Dash model. To construct the corresponding Dot model, we add two lightweight down-sampling blocks and up-sampling blocks on the top and under the bottom of each pre-trained Dash model respectively. With the weight sharing strategy, all the Dot models are trained following the official training settings. As shown in Table 7, we provide the detailed information, including the source models, the speeds of the Dash models and Dot models and $N$. Recall that the Dot model is $N$ times faster than the Dash model. All the speeds for different models are tested using a single NVIDIA GeForce RTX 3090. When testing the speeds, we set the batch size to 100 for most benchmarks except 20 for CelebA-HQ and MS-COCO benchmarks. During the training procedures, a Dot model is trained to estimate the difference between the outputs from the Dash model at two randomly sampled steps. Here, we give an example of the training procedure and sampling procedure for DDIM sampler, as shown in Algorithm 1 and Algorithm 2. The procedures can be easily extended to other samplers with simple modification. The Dot models are trained on the servers with 8 NVIDIA

Tesla V100 GPUs or 8 NVIDIA GeForce RTX 4090 GPUs.

$N$ **on Different GPUs.** Recall that Dot is $N$ times faster than Dash. For a Dash model and its trained Dot model, the speedup of Morse gets larger when $N$ gets larger. While the speeds of the models may vary on different GPUs, leading to the change of $N$ and speedup. In our design, we construct a Dot model by adding several extra blocks on the top and under the bottom of the pre-trained Dash model. Therefore, a Dot model has the architecture which is very similar with its corresponding Dash model. As shown Table 7, we can find that a pair of Dash and Dot mostly has little change in $N$ on different GPUs. The results indicate that our Morse performs well on different GPUs.

---

**Algorithm 1:** Training of Dot with DDIM

**Require :** Trained Dash model $\theta(\cdot, \cdot)$
**Require :** Dot model $\eta(\cdot, \cdot, \cdot, \cdot, \cdot)$ to be trained
**Require :** Schedule function $\phi(\cdot, \cdot, \cdot, \cdot)$
**Require :** Dataset $\mathcal{D}$
**Require :** Learning rate $\gamma$

1 **repeat**
2     sample $\mathbf{x} \sim \mathcal{D}$
3     sample $\epsilon \sim \mathcal{N}(0, \mathbf{I})$
4     sample $t_s, t_o \sim U[0, T]$ $(t_s > t_o)$
5     $\mathbf{x}_{t_s} = \alpha_{t_s}\mathbf{x} + \sigma_{t_s}\epsilon$
6     $\mathbf{z}_{t_s} = \theta(\mathbf{x}_{t_s}, t_s)$
7     $\mathbf{x}_{t_o} = \phi(\mathbf{x}_{t_s}, \mathbf{z}_{t_s}, t_s, t_o)$
8     $\mathbf{z}_{t_o} = \theta(\mathbf{x}_{t_o}, t_o)$
9     $\hat{\mathbf{z}}_{t_o} = \mathbf{z}_{t_s} + \eta(\mathbf{x}_{t_s}, \mathbf{x}_{t_o}, \mathbf{z}_{t_s}, t_s, t_o)$
10    $\eta \leftarrow \eta - \gamma \nabla_\eta \|\mathbf{z}_{t_o} - \hat{\mathbf{z}}_{t_o}\|_2^2$
11 **until** *convergence*

---

**Algorithm 2:** DDIM Sampling with Morse

**Require :** Trained network Dash $\theta(\cdot, \cdot)$
**Require :** Trained network Dot $\eta(\cdot, \cdot, \cdot, \cdot, \cdot)$
**Require :** Schedule function $\phi(\cdot, \cdot, \cdot, \cdot)$
**Require :** Sequence of time points $t_n > t_{n-1} > \cdots > t_0$
**Require :** Number of dash steps $d$

1 sample $\mathbf{x}_{t_n} \sim \mathcal{N}(0, \mathbf{I})$
2 uniformly sample $s_d, \ldots, s_0$ from $t_n$ to $t_0$
3 **for** $i \leftarrow n$ **to** 1 **do**
4     **if** $t_i \in \{s_d, \ldots, s_1\}$ **then**
5        $\mathbf{z}_{t_i} = \theta(\mathbf{x}_{t_i}, t_i)$
6        $t_s = t_i$
7     **else**
8        $\mathbf{z}_{t_i} = \mathbf{z}_s + \eta(\mathbf{x}_{t_s}, \mathbf{x}_{t_i}, \mathbf{z}_{t_s}, t_s, t_i)$
9     **end**
10    $\mathbf{x}_{t_{i-1}} = \phi(\mathbf{x}_{t_i}, \mathbf{z}_{t_i}, t_i, t_{i-1})$
11 **end**
**Return :** $\mathbf{x}_{t_0}$

---

**How to Determine the Scheduling of Morse.** Generally, how the scheduling of Morse is determined relies on three key factors: **(1)** The selection of the jump-sampling-based diffusion sub-sequence of the Dash model. With jump sampling, only a sub-sequence of the available time steps (e.g., T,...,0) are visited. Clearly, there are different strategies to select the sub-sequence. In Morse, we typically use a simple selection strategy following the standard jump sampling settings (which are supported with most existing diffusion methods) of any pre-trained diffusion model (i.e. the Dash model), for easy implementation; **(2)** Given the numbers of sampling steps for Dash and Dot, how to decide their orders. For a diffusion process with Morse, we denote the number of sampling steps for Dash as $k_{dash}$ and the number of sampling steps for Dot as $k_{dot}$, leading to $k_{dash} + k_{dot}$ sampling steps in total. Since the Dot model needs the trajectory information of the Dash model for predicting its residual feedback, the first sampling step can only be with the Dash model, and then we can flexibly decide orders of the remaining steps. For simplicity, we uniformly sample the steps with the Dash and Dot models. For example, if $k_{dash} = 2$ and $k_{dot} = 4$, the order of the steps can be denoted as $Dash, Dot, Dot, Dash, Dot, Dot$; **(3)** Given the desired latency budget, how to decide the number of sampling steps for Dash and Dot. Recall that Dot is $N \times$ faster than Dash. Under the desired latency of $n$ LSDs (LSD is the defined time metric, namely the time for the baseline diffusion model to perform one step), there could be $n - k$ $(0 \leq k < n)$ sampling steps with Dash and $Nk$ sampling steps with Dot for Morse. The problem is how to decide the $k$. Under ideal conditions where Dot and Dash perform exactly the same for noise estimation, Morse achieves a speedup of $(n - k + Nk)/n$. While Morse can accelerate a pre-trained diffusion model with a wide range of $k$ (as shown in Fig. 13). Based on our experiments with 10 diffusion models on 6 benchmarks, we suggest setting $(n - k + Nk)/n$ between 2.0 and 3.0, which leads mostly to the best results. With the above three simple principles, it's easy and flexible to decide the scheduling of Morse under different latencies.

### A.5. More Experiments for Studying Morse

**Morse with Different Samplers.** In the experiments described in Sec. 3.2, we evaluate our Morse on CIFAR-10 ($32 \times 32$) benchmark with different samplers. Here, we perform experiments to further validate the effectiveness of Morse on other datasets with different samplers. As shown in Fig. 11, our Morse achieves good generalization ability on CelebA-HQ ($256 \times 256$) dataset with different samplers.

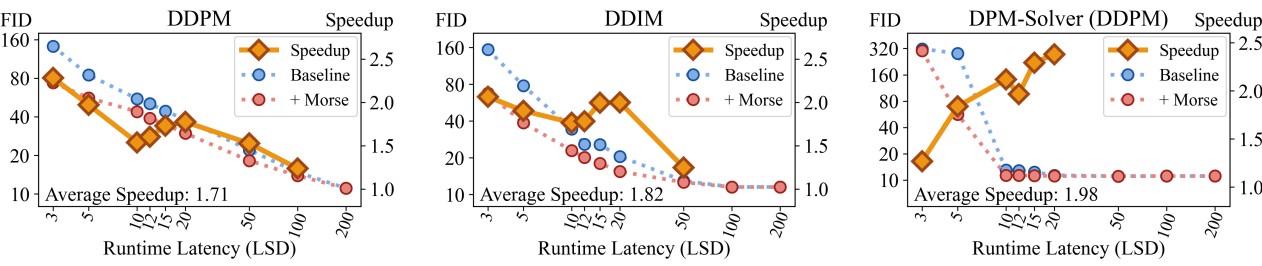

*Figure 11.* Results of Morse with different samplers on CelebA-HQ (256×256) benchmark.

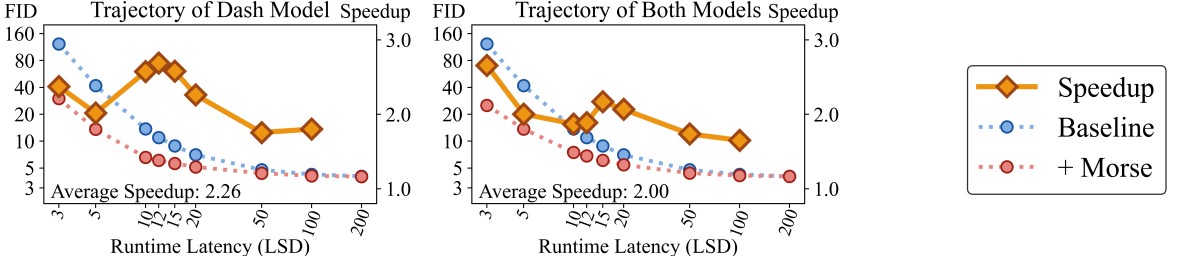

*Figure 12.* Results of Dot with trajectory information from the Dash model and the both two models.

**Where Trajectory Information Comes from?** Recall that Morse redefines how to estimate noise during the generation process as:

$$\mathbf{z}_{t_i} = \begin{cases} \theta(\mathbf{x}_{t_i}, t_i) & t_i \in S \\ \mathbf{z}_{t_s} + \eta(\mathbf{x}_{t_i}, \mathbf{x}_{t_s}, t_i, t_s, \mathbf{z}_{t_s}) & t_i \notin S \end{cases}. \tag{6}$$

In the design, a Dot model generates residual feedback conditioned on the observations at the current JS point on the trajectory of the Dash model. For another reasonable design, the observations can also come from the trajectory of the two models, which can be represented as:

$$\mathbf{z}_{t_i} = \begin{cases} \theta(\mathbf{x}_{t_i}, t_i) & t_i \in S \\ \mathbf{z}_{t_{i-1}} + \eta(\mathbf{x}_{t_i}, \mathbf{x}_{t_{i-1}}, t_i, t_{i-1}, \mathbf{z}_{t_{i-1}}) & t_i \notin S \end{cases}. \tag{7}$$

In the experiments, we compare the two designs which utilize the different trajectory information on CIFAR-10 dataset with DDIM sampler. The results are shown in Fig. 12. It can be seen that our design (using trajectory information from the Dash model) performs better, which achieves an average speedup of $2.26\times$ against to $2.00\times$. In which case the number of steps is extremely small (e.g., 3), using trajectory information from $t_{i-1}$ is better than that from $t_s$. This is probably because the distance between $t_s$ and $t_i$ becomes relatively large when the number of steps is very small, which makes the trajectory information less helpful for the Dot model. We can also find that the Dot model also works well with the trajectory information from itself, though it is trained with the trajectory information from the Dash model during the training procedure.

**Comparison under the Same Number of Steps.** For evaluating the speedups of Morse under different sampling step budgets, we mostly compare the DMs with and without Morse under the selected latencies in the previous experiments. In Fig. 14, we provide the results of DMs under the selected number of steps, establishing a set of different time-interleaved configurations of Dot and Dash under a given LSD budget. It can be seen that the curves of a DM with Morse are always below the curve without Morse, indicating the consistent acceleration ability of Morse under different numbers of sampling steps and proportion between the total numbers of sampling steps and the numbers of sampling steps with noise estimation from Dot.

**Effect of Exchanged Steps Ratio.** Recall that under a specific latency of $n$ LSDs, there could be $n - k$ ($0 \le k < n$) sampling steps with Dash and $Nk$ sampling steps with Dot for Morse. Morse can flexibly change the JS step length by controlling $k$. Here, we define the ratio of exchanged steps as $k/n$. In the experiments, we explore the effect of different

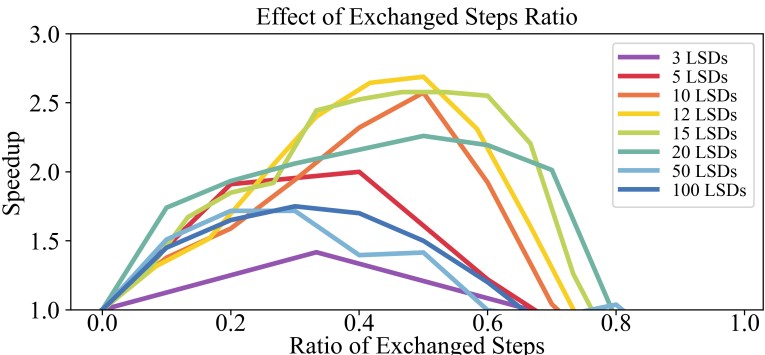

*Figure 13.* Speedups of Morse with DDIM sampler on CIFAR-10 (32×32) under different LSDs and exchanged steps ratios. The exchanged steps ratio denotes the ratio of the latency of steps with Dot to the total latency in a generation process.

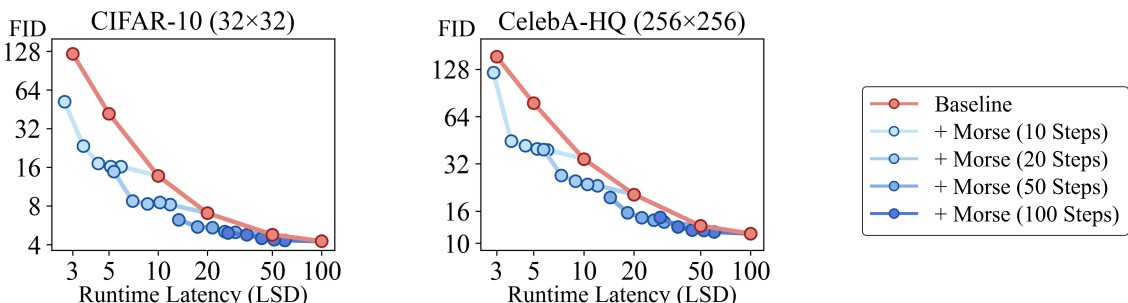

*Figure 14.* Results of Morse with DDIM sampler under different numbers of sampling steps. For a diffusion process with Morse, we set 50%, 60%, 70%, 80% and 90% of the sampling steps for using the Dot model and the other steps using the Dash model.

ratios of the exchanged steps. We conduct the experiments on CIFAR-10 dataset with DDIM sampler. The results are shown in Fig. 13. Under most LSDs, Morse can achieve a speedup around 2× with most ratios. Under the extreme condition when we exchange most of the sampling steps with Dash for the sampling steps with Dot (e.g., more than 70%), the speedups sharply decrease below 1.0×. In our opinion, the reason is that the trajectory information becomes less helpful for the Dot models on residual estimation when the distance between the two sampling steps is very large, since there are much fewer sampling steps with Dash.

**Different Designs for the Extra Down-Sampling and Up-Sampling Blocks of Dot.** Recall that we construct the Dot model for Stable Diffusion by adding 2 trainable lightweight down-sampling blocks and up-sampling blocks to the Dash model. When training the Dot model, we fix the shared pre-trained layers and adopt lightweight LoRA. In the experiments, we study the construction of a Dot model with different designs for down-sampling and up-sampling. We evaluate several variant designs for the Dot model including: (1) Down-sampling and up-sampling with proposed trainable blocks or bilinear sampling; (2) Training the Dot model with or without LoRA. The shared pre-trained layers are fixed. From the results shown in the Table 8, we can see that both designs can significantly improve the performance of Morse. Without fine-tuning, the original DM cannot adapt well to a lower resolution directly. While adding the trainable sampling blocks and adopting LoRA for training the Dot model can enhance learnable and soft resolution transformation and help the pre-trained blocks adapt to the new resolutions, respectively.

**Different Architectures of the Dot Model.** In the experiments, we evaluate the performance of Morse with the independent Dot model without sharing the pre-trained blocks with Dash model. We conduct the experiments on MS-COCO dataset with Stable Diffusion. In the variant design, the Dot model has similar architecture with the Dash model while with the reduced numbers of channels and blocks. The results are shown in Table 9. Even without fine-tuning with the pre-trained weights, our Morse can still accelerate the Stable Diffusion. While it needs more training iterations and trainable parameters to achieve similar performance with our proposed design. The results demonstrate the effectiveness and efficiency of our proposed weight sharing strategy for training a Dot model. While it also shows that we can flexibly construct a Dot model with different architectures.

*Table 8.* FIDs of Stable Diffusion with different variants of Dot. We calculate FIDs under different classifier-free guidance scales and select the best FID among all the scales.

| Method | Trainable Sampling Blocks | LoRA | 10 LSDs | 15 LSDs | 20 LSDs | 50 LSDs |
|---|---|---|---|---|---|---|
| Stable Diffusion | - | - | 10.65 | 9.47 | 8.70 | 8.22 |
| + Morse | | | 370.79 | 397.82 | 392.64 | 389.12 |
| | ✓ | | 9.23 | 8.94 | 8.60 | 8.36 |
| | | ✓ | 9.79 | 9.21 | 8.89 | 8.51 |
| | ✓ | ✓ | 8.60 | 8.55 | 8.29 | 8.15 |

*Table 9.* Different architectures of the Dot model for Stable Diffusion.

| Method | Training Iterations | Params | Average Speedup |
|---|---|---|---|
| Independent Dot model | 0.4 million | 324.93M | 2.07× |
| Dot model with weight sharing strategy | 0.1 million | 97.84M | **2.29×** |

## A.6. More Generated Samples

We provide some generated samples from the diffusion models with and without Morse under different LSDs for better comparisons, including image generation for CelebA-HQ (256×256) and LSUN-Church (256×256), text-to-image generation on MS-COCO with Stable Diffusion v1.4 and LCM-SDXL.

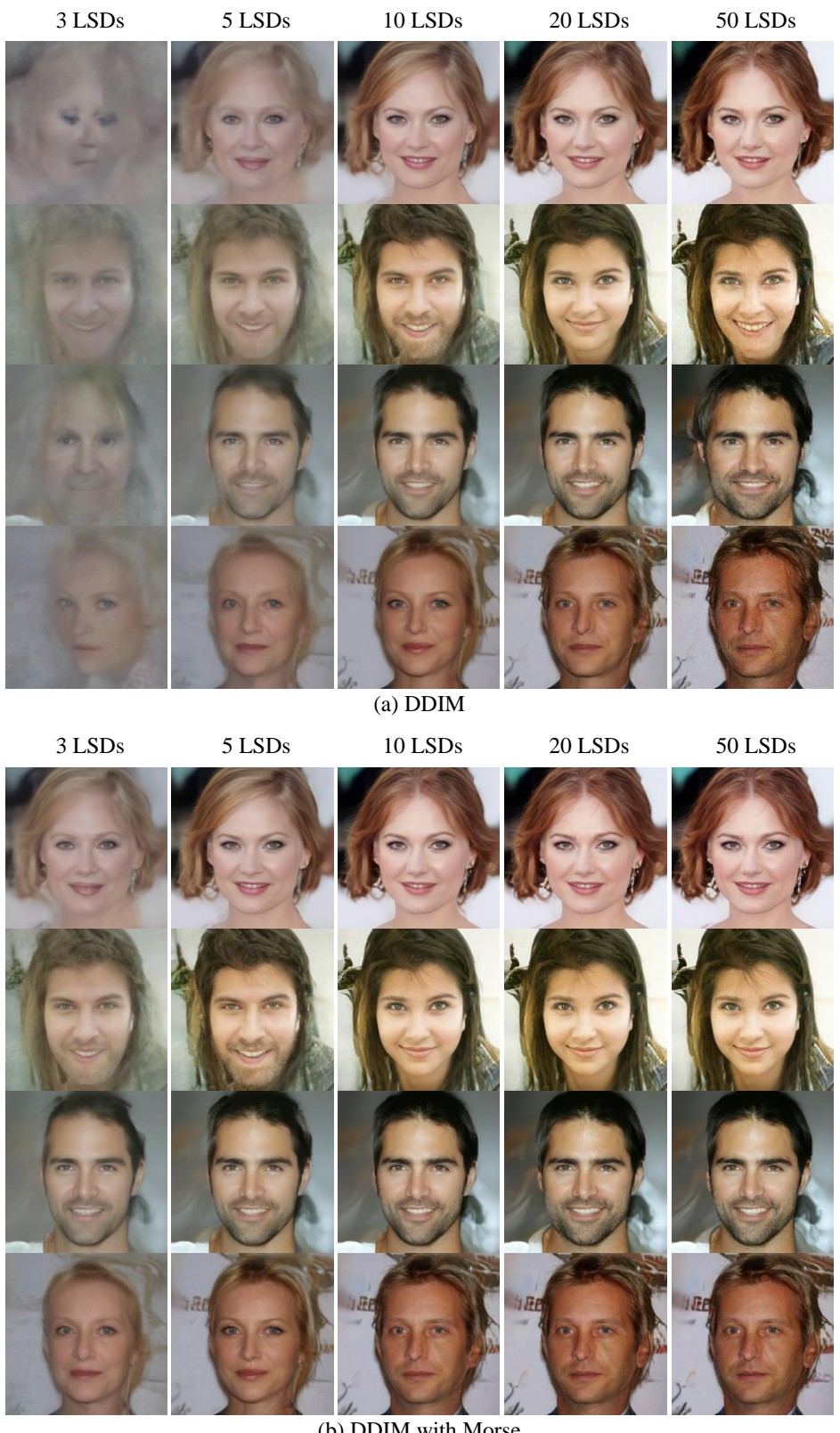

Figure 15. Generated samples at resolution 256×256 for CelebA-HQ dataset using DDIM sampler with and without Morse.

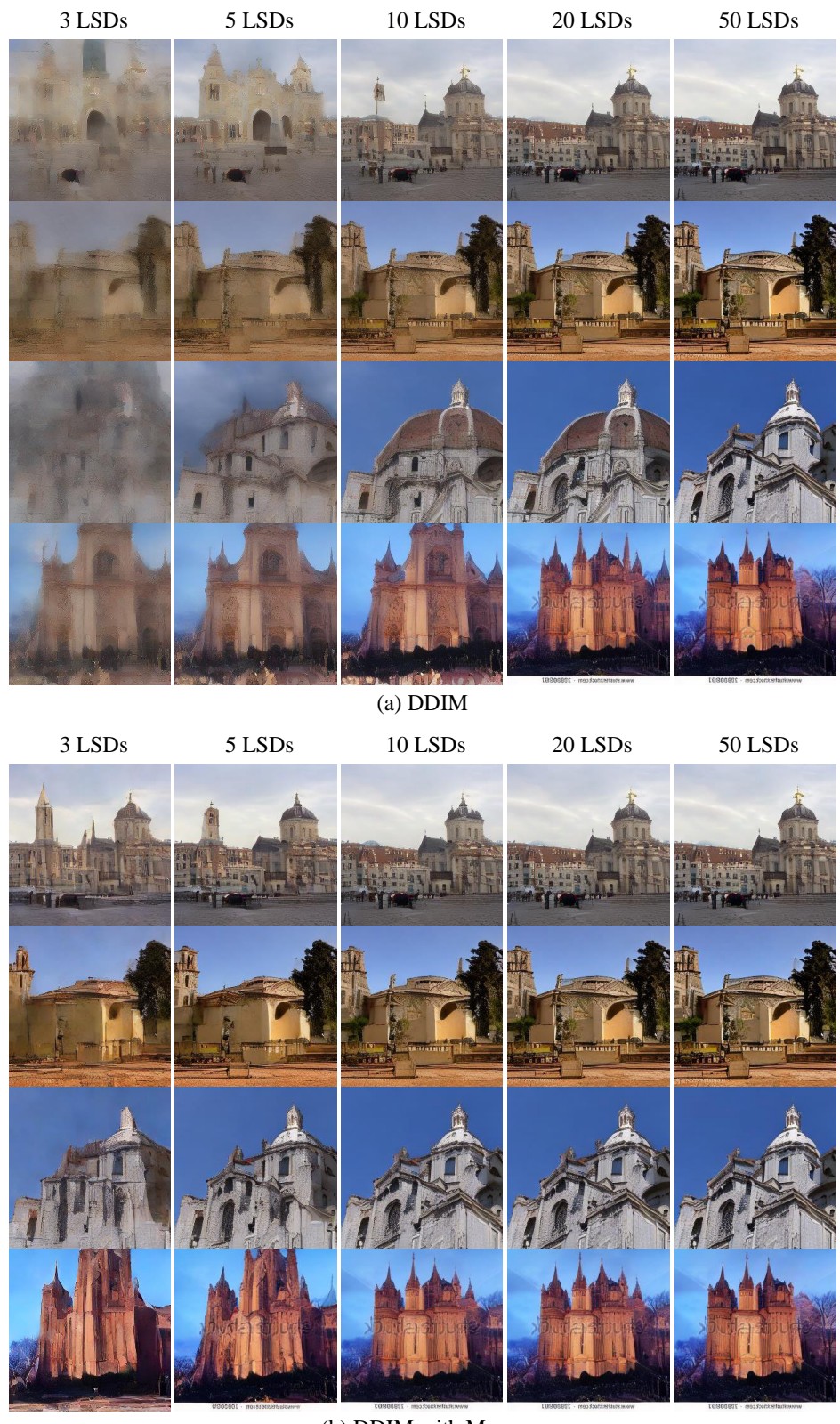

Figure 16. Generated samples at resolution 256×256 for LSUN-Church dataset using DDIM sampler with and without Morse.

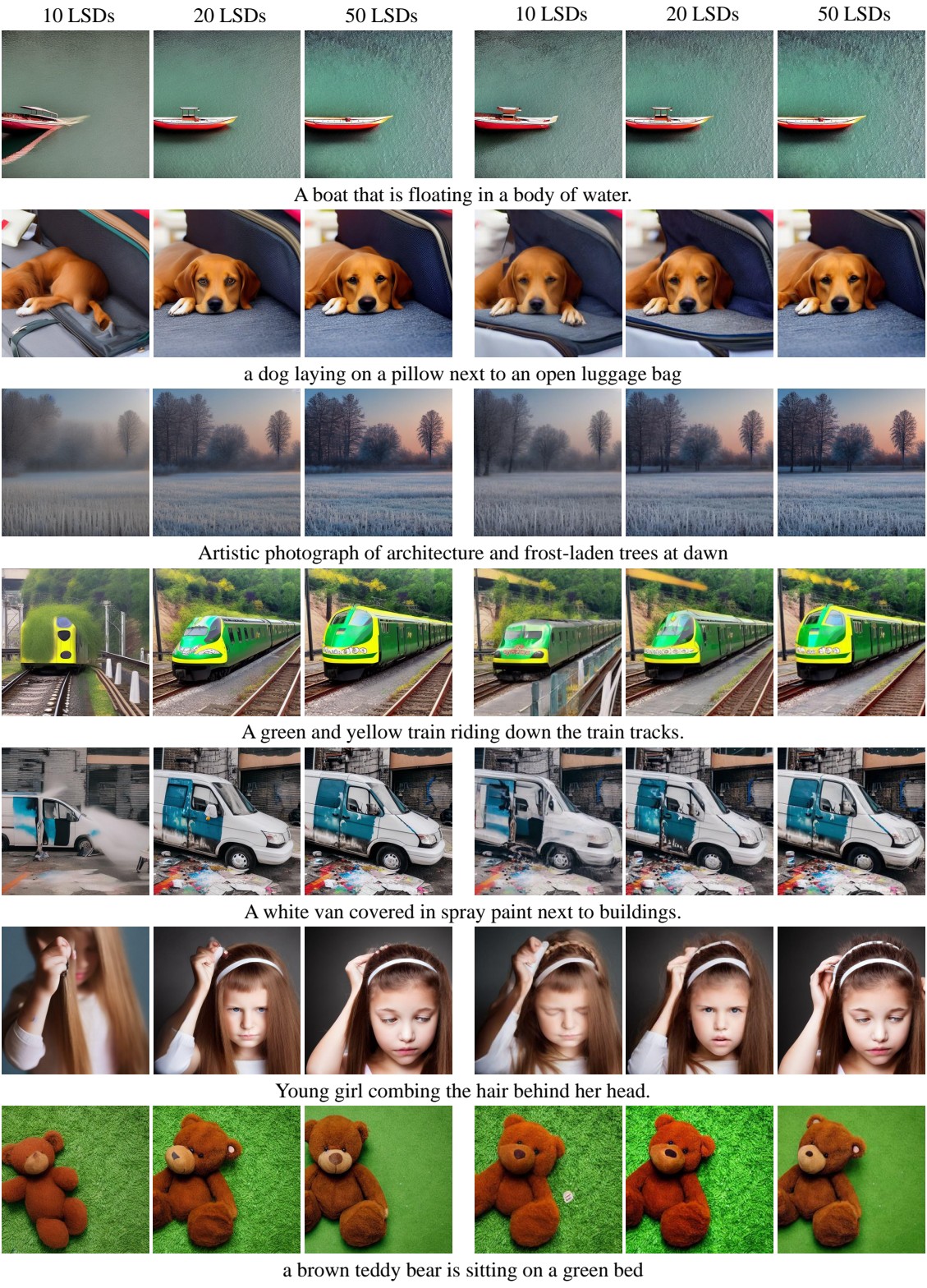

Figure 17. Generated samples at resolution 512×512 with prompts from MS-COCO validation set from Stable Diffusion v1.4 using DDIM sampler with and without Morse. The classifier-free guidance scale is set to 7.5 following the official settings.

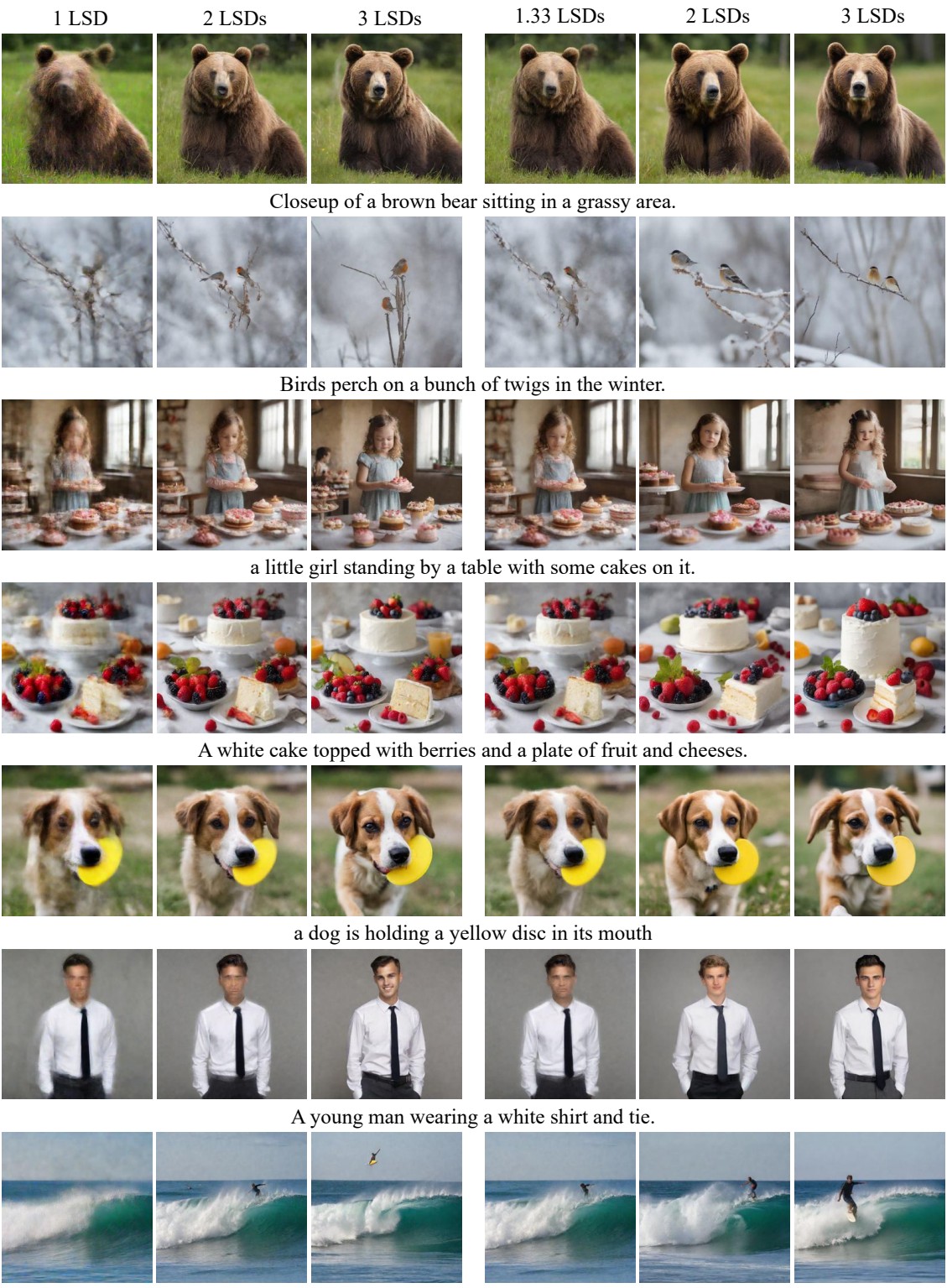

Figure 18. Generated samples at resolution 1024×1024 with prompts from MS-COCO validation set from LCM-SDXL with and without Morse.

