# OpenReview forum: "Morse: Dual-Sampling for Lossless Acceleration of Diffusion Models"
_ICML.cc/2025/Conference — ICML 2025 poster_

### Official Review · Reviewer_cEcd · 2025-03-15

**Overall Recommendation:** 3

**Summary:**

This work proposes a method to speed up diffusion models by skipping steps (Dash) and by compensating errors for the Dash (Dot). The Dash model uses a pre-trained model with mere skipping steps (i.e., existing diffusion models such as Stable Diffusion, but using larger, less steps) while the Dot model must be trained to provide adaptive residual feedback. LoRA structure was used for the Dot model along with the weight sharing with the Dash model for efficient tuning. This work was tested with Stable Diffusion and LCM-SDXL, showing 1.78x to 3.31x speed ups with comparable performance.

**Claims And Evidence:**

While there were some unclear aspects, this manuscript seems to provide partial evidence for their claims such as speed ups (1.78x - 3.31x), thus demonstrating the effectiveness of the proposed method. However, these claims were not well-compared with other recent works, so it is not straightforward to see if this proposed method is indeed a state-of-the-art. In fact, it is unclear if the proposed method is indeed "fast" as claimed over the entire manuscript since there are a number of missing comparisons and a number of new works on one-step diffusion models. Moreover, "Universally" in the title seems an over-claim since the proposed architecture of the Dot model was not tested universally (enough) for other recent diffusion models.

**Essential References Not Discussed:**

As mentioned above, the following papers and other recent works should be properly discussed and compared:
- X Ma et al., Deepcache: Accelerating Diffusion Models for Free, CVPR 2024 (cited, but not discussed)
- G Wang et al., PFDiff: Training-free Acceleration of Diffusion Models Combining Past and Future Scores, ICLR 2025 (not cited)

**Experimental Designs Or Analyses:**

Experimental designs or analyses were good except for the lack of comparisons with other similar baselines such as DeepCache (or PFDiff). While they look different from the proposed method, they were also using the temporal redundancies of diffusion steps in different ways (and without tuning unlike the proposed method, which seems important), so it is important to clearly show the advantage of the proposed method over related prior arts using the same underlying principles.

**Methods And Evaluation Criteria:**

The methods make sense to me. However, Table 2 looks like somewhat unfair comparisons since the Stable Diffusion is the pre-trained model and this work is the tuning model. There are a number of recent works that do not even require tuning such as Ma et al. CVPR 2024 (cited in this manuscript) and Wang et al. PFDIff, ICLR 2025 (available online since last year), so this table seems to deliver a biased message without properly discussing with other related works.

**Other Comments Or Suggestions:**

In Figure 1 caption, "Latency Consistency Models" should be "Latent Consistency Models"

**Other Strengths And Weaknesses:**

While this work seems to propose a clear method for accelerating diffusion models, there are a number of weaknesses as below:
- Requiring additional training for speed up does not look attractive over other training-free methods (PFDiff, Deepcache) or one-step diffusion models. For a new model, this work requires an additional training (190 A100 hours may not be much as compared to the original model training, but can be cumbersome when a new model is given multiple times). It is also unclear how this work can work with personalization of diffusion models or concept erasing of diffusion models.
- As mentioned above, this work seems to demonstrate its effectiveness in Stable Diffusion Models only, but there are a number of other diffusion models that are available online. It seems important to demonstrate for some of them, especially to justify the Dot method.

**Questions For Authors:**

In Figure 8, it was interesting to see that the major FID improvements were done around the cases with CLIP scores near 28-29, while the improvements were almost none for the lowest CLIP score cases (around 25) and the highest CLIP score cases (around 30). Any insight for them?

What was the unit of LSD? In Figures 4-5, there were large FID gaps around 3-5 LSDs for the baseline and the proposed method, but their runtime latencies were at the same point - in terms of absolute computational time scales, they should not be exactly matching due to additional computation of the Dot model, right? Any explanation for them? Actually, these results were also my major concerns since even 3-5 LSD with Morse did not achieve great results as compared to consistency models with around 4 steps (not compared in here) or more recent one-step models.

Will this work (and other recent works such as Deepcache) be useful for one-step diffusion models?

**Relation To Broader Scientific Literature:**

Step skipping strategies while exploiting temporal redundancies in diffusion steps have been discussed and developed such as DeepCache or PFDiff, but this work unfortunately did not properly address and compare with them in my view. Moreover, there are a number of recent works on one step diffusion models, so the insight in this work may become no so useful anymore quickly as the major diffusion models vendors will adopt this one step approaches. Full discussion on these other works and potential limitation seem important to justify the key contributions of this work.

Here are some examples for step skipping methods in diffusion models:
- Wenyang Zhou et al., EMDM: Efficient Motion Diffusion Model for Fast and High-Quality Motion Generation, ECCV 2024.
- Hui Zhang et al., AdaDiff: Adaptive Step Selection for Fast Diffusion Models, AAAI 2025.
- Zhenyu Zhou et al., Fast ODE-based Sampling for Diffusion Models in Around 5 Steps, CVPR 2024.

It is especially important to see Zhou et al. CVPR 2024 since this work is about using large step sizes with the aid of corrections.

**Theoretical Claims:**

There is no theoretical claim in this manuscript as far as I know.

---

> ### Author Rebuttal · Authors · 2025-03-31
>
> Thanks a lot for your detailed comments and the recognition of our method.
>
> **1. To your main concern** about comparisons of Morse with recent related methods, **our responses include 4 parts**:
>
> **Part 1: Comparison with DeepCache and PFDiff.** In formulation, they and our Morse explore the temporal step redundancies for accelerating diffusion models in different ways: **1)** DeepCache (DC, for brevity) reuses the features at step t for N-1 following steps, and PFDiff utilizes the states of past steps stored in a buffer to update the states of future steps, and they are training-free; **2)** Morse is a simple dual-model design relying on step skipping and residual feedback with a small number of learnable parameters. In performance: **1)** Table A compares Morse with open-source DC on CIFAR-10 under the same settings. We can see: **a)** DC is lossy in output quality while Morse is lossless; **b)** Morse performs better in terms of both FID and Throughput; **2)** In Part A of our responses to Reviewer vMF4, we provide more results on Stable Diffusion (SD), further showing the superiority of Morse to DC; **3)** Table B compares Morse with PFDiff (code is public) on SD, showing Morse is better than PFDiff.
>
> **Table A**: Results on CIFAR-10. **DC denotes DeepCache**
> |Method|FID↓|Throughput↑(img/second on a 4090 GPU)|
> |-|-|-|
> |DDIM(100 steps)|4.3|14.4|
> |+DC(N=2, N: the cache updating at every N steps)|4.4|19.1|
> |+DC(N=3)|4.8|22.2|
> |+DC(N=5)|5.7|23.7|
> |+DC(N=10)|10.0|27.2|
> |+Morse(LSD=100)|**4.1**|14.4|
> |DDIM(50 steps)|4.8|29.1|
> |+Morse(LSD=50)|4.4|28.8|
> |DDIM(20 steps)|7.0|71.7|
> |+Morse(LSD=20)|5.1|**73.9**|
>
> **Table B**: Results on MS-COCO. Evaluated on 10k samples.
>
> |Method\LSDs|5|10|15|20|
> |-|-|-|-|-|
> |SD|23.9|16.8|16.1|16.0|
> |+PFDiff|18.3|13.1|13.6|14.0|
> |+Morse|**13.9**|**11.7**|**12.2**|**13.6**|
>
> **Part 2: Comparison with Step Skipping Methods.** Thanks for pointing out three works in this line: **1)** EMDM is a GAN-based diffusion learning method tailored for video-based human motion generation, and AdaDiff is an RL-based diffusion learning method tailored for text-to-image generation in a prompt-difficulty-aware manner. Our Morse addresses a variety of image generation tasks (see Sec.3), thus it differs with them in focus and formulation (see Part 1); **2)** AMED (Zhou et al., CVPR 2024) is an improved solver using a 2-MLP network for each steps-budget to learn the mean direction for fast sampling. Table C compares Morse with AMED on CIFAR-10, showing Morse is better; **3)** In Part B of our responses to Reviewer vMF4, we provide more experiments, showing Morse is also better than other step skipping methods.
>
> **Table C**: Results on CIFAR-10
> |Method\LSDs|3|5|7|9|
> |-|-|-|-|-|
> |DPM-Solver++|110.0|24.97|6.74|3.42|
> |+AMED|25.95|7.68|4.51|3.03|
> |+Morse|**14.36**|**5.73**|**3.31**|**2.98**|
>
> **Part 3: Comparison with One-step Diffusion Models.** They aim to get the extreme diffusion speedup, yet lead to serious degeneration of output quality compared to full-step teachers. Methods based on step skipping (e.g., Morse & AMED) and past-step feature reuse (e.g., DeepCache & PFDiff) cannot accelerate one-step diffusion models that have no temporal redundancy, which is a common limitation. To avoid an over-claim, "universally" will be removed in our paper.
>
> **Part 4:** Morse is a leading lossless method that needs to train only once given a pre-trained diffusion model. "190 A100 hours" is for a billion-level SD model (its pre-training cost is 1000x longer, see Table 2). For more comparisons, please check [this link](https://anonymous.4open.science/r/Morse-Reviewer-cEcd) and our responses to Reviewer vMF4/dKeM.
>
> **2. To Q1** regarding Fig.8, the FID improvements are decent to the highest CLIP score cases, but are really small to the lowest cases, as illustrated in Table D below. This is due to the random sampling strategy of guidance scales (GS) from 2 to 10 when training the Dot model (see Line580-585). The lowest CLIP scores appear at GS=2 occupying a small portion of samples, making it hard to improve FIDs.
>
> **Table D:** Results (FID|CLIP score) on MS-COCO with SD
> |Method\LSDs|10|20|50|
> |-|-|-|-|
> |SD(GS=2)|18.8\|25.4|11.1\|26.5|9.0\|27.0|
> |+Morse|18.7\|25.4|11.1\|26.5|9.0\|27.0|
> |SD(GS=7.5)|11.8\|29.7|12.4\|30.1|13.5\|30.0|
> |+Morse|9.3\|29.8|11.3\|30.1|13.3\|30.0|
>
> **3. To  Q2** regarding what is the unit of LSD, it is the Latency per Step of the baseline Diffusion model, i.e., the one-step time cost of the Dash model (see Line244-253). Recall that, in our Morse, Dot is N times faster than Dash, so a diffusion process with n-k Dash steps (skipping k steps totally) and Nk Dot steps has the same latency (n LSDs) to the baseline running n steps. This is why, given n LSDs, the runtime latencies of the diffusion model with and w/o Morse are at the same point in Fig.4-5. The runtime speed under different settings of n LSDs is shown in Table 5 of our paper.
>
> **4. To  Q3**, the answer is no, as clarified in Part 3.

---

### Official Review · Reviewer_dKeM · 2025-03-15

**Overall Recommendation:** 4

**Summary:**

This paper proposes a new faster sampling method for diffusion models. The goal of the paper is to provide a faster sampling method, not sacrificing performance (unlike many recent distillation-based methods). The main idea is to use another (the "dot") model in the sampling process. This additional sampler "corrects" the jump sampling of the main model, and is relatively quick and requires less training effort (downsizing a pretrained model and adding minimal extra layers, trained by LoRA). This enables more runs in the same time period, which greatly enhances the sampling efficiency. Accordingly, the focus boils down to reducing the entire time cost rather than minimizing the number of steps. Experiments show that the proposed method can enhance the performance of existing models and provide a two- to four-times speed up.

## update after rebuttal

Some reviewers pointed out that using an additional model can be a burden, and it might not be as favorable as other competing approaches. However, I still think that the two-denoiser approach is interesting and novel, and the additional results in the rebuttal sufficiently eliminate the doubts. I maintain my original score.

**Claims And Evidence:**

Using another (lightweight) sampler to improve the sampling process in diffusion models is quite new and interesting. Recent one-step generators are somewhat limited in performance, but this method can be a viable alternative without sacrificing performance. The experimental results support this claim.

**Essential References Not Discussed:**

The bibliography is thorough enough.

**Experimental Designs Or Analyses:**

The paper provides extensive ablation (and hyperparameter tuning) experiments. The results are generally convincing.

However, one complaint I have is that there is no direct comparison to distillation-based methods. I do understand that this paper's focus is somewhat different from these, but as the paper criticizes their shortcomings, comparing the performance/speed to these will provide a better picture.

**Methods And Evaluation Criteria:**

The design of the "dot" model is quite simple and efficient. The proposed method is evaluated on Stable Diffusion and LCM-SDXL, which is good enough in my opinion.

**Other Comments Or Suggestions:**

As mentioned above, comparing the performance/speed to the distillation-based method will benefit the readers.

**Other Strengths And Weaknesses:**

Please see the above points.

**Questions For Authors:**

- In (5), the "dot" model receives five inputs, three of which are images. This model is a modified version of a pretrained diffusion model, so how are these inputs combined? Are they concatenated, and then is the resulting channel reduced by the first (extra) layer? This is not really explained in the paper or the supplementary material.

- Moreover, the explanation about the structure of the dot model itself is somewhat confusing. It says that the pretrained model is fixed, two extra layers are added, and then LoRA is applied? This was somewhat confusing, and after seeing Fig. 3, I guessed that LoRA is applied to the pretrained part. Articulating the description would benefit the readers.

**Relation To Broader Scientific Literature:**

Diffusion models have become a significant part of generative models, so providing a faster method without sacrificing performance can benefit many related areas. Moreover, the main concept (using another auxiliary sampler to correct jump sampling) is quite novel and interesting.

**Theoretical Claims:**

N/A

---

> ### Author Rebuttal · Authors · 2025-03-31
>
> Thank you so much for the constructive comments, and the recognition of our work including the proposed method, the experiments and the performance. Please see our below responses to your concerns and questions one by one.
>
> **1 To your concern about** “However, one complaint I have is that there is no direct comparison to distillation-based methods…comparing the performance/speed to these will provide a better picture.”
>
> **Our responses are:** Thanks for your valuable suggestion. We agree that a direct comparison of Morse with distillation-based methods would improve our work further.
> In Section 3.4 of our paper, we have shown that Morse can be combined with consistency distillation, which achieves an average speedup 1.43× for Latent Consistency Model on MS-COCO (see Table 3).
>
> Here, we further provide a direct comparison with **BK-SDM** [1], a state-of-the-art distillation method that constructs lightweight diffusion models through architectural compression and feature distillation. As shown in Table A, when comparing Stable Diffusion+Morse (10 LSDs) with BK-SDM-base|-small|-tiny models (25 steps), Morse achieves better FID scores (8.60 vs 13.35|14.17|15.60) while maintaining higher throughput (4.67 vs 2.79|2.84|2.92), demonstrating the superiority of Morse to BK-SDM.
>
> Additionally, following the suggestions from Reviewer vMF4/cEcd, we provide further comparisons with other acceleration paradigms (e.g., feature reuse, sampling schedule optimization, quantization). For instance, our Morse outperforms DeepCache [2], a state-of-the-art feature reuse method, on MS-COCO with Stable Diffusion. The results are presented in Table B below. More results and discussions can be found in our responses to Reviewer vMF4/cEcd.
>
> **Table A**: Comparison of Morse and BK-SDM on MS-COCO with DDIM solver. The results are measured with the same settings as in our paper. Throughput is measured on a single RTX 4090 GPU. BK-SDM-base, BK-SDM-small and BK-SDM-tiny are student networks with different architectures distilled by Stable Diffusion as the teacher network. LSD denotes the Latency per Step of the baseline Stable Diffusion.
>
> |Method|FID↓|Throughput↑ (images/second)|
> |-|:-:|:-:|
> |BK-SDM-base (25 steps)|13.35|2.79|
> |BK-SDM-small (25 steps)|14.17|2.84|
> |BK-SDM-tiny (25 steps)|15.60|2.92|
> |Stable Diffusion (25 LSDs) |8.41|1.95|
> |+ Morse (25 LSDs)|**8.16**|1.97|
> |Stable Diffusion (20 LSDs)|8.70|2.45|
> |+ Morse (20 LSDs)|8.29|2.43|
> |Stable Diffusion (10 LSDs)|10.65|4.62|
> |+ Morse (10 LSDs)|8.60|**4.67**|
>
> **Table B**: Results of Morse and DeepCache on MS-COCO with Stable Diffusion v1.4 and DDIM solver. N denotes the cached feature maps of DeepCache is updated at every N steps.
>
> |Method|FID↓|Throughput↑ (images/second)|
> |-|:-:|:-:|
> |Baseline (50 steps)|8.22|1.07|
> |Baseline (20 steps)|8.70|2.45|
> |Baseline (10 steps)|10.65|4.62|
> |DeepCache (50 steps, N=2)|9.31|1.62|
> |DeepCache (50 steps, N=3)|9.55|1.94|
> |DeepCache (50 steps, N=5)|10.62|2.33|
> |Morse (50 LSDs)|**8.15**|1.12|
> |Morse (20 LSDs)|8.29|2.43|
> |Morse (10 LSDs)|8.60|**4.67**|
>
> [1] Kim, Bo-Kyeong, et al. "Bk-sdm: A lightweight, fast, and cheap version of stable diffusion." ECCV, 2024.
>
> [2] Ma, Xinyin, Gongfan Fang, and Xinchao Wang. "Deepcache: Accelerating diffusion models for free." In CVPR, 2024.
>
> **2. To your question about** “In (5), the "dot" model receives five inputs, three of which are images… Are they concatenated, and then is the resulting channel reduced by the first (extra) layer?…”
>
> **Our responses are:**  As shown in Figure 2 of our paper, the Dot model receives five inputs, including three image features ($x_{t_{i}}$, $x_{t_{s}}$ and $z_{t_{s}}$) and two scalars ($t_{i}$ and $t_{s}$). As you noted, the three image features are concatenated before being fed into the first extra layer. The two scalars ($t_{s}$ and $t_{i}$), corresponding to the time steps of Dash and Dot respectively, are individually embedded and then concatenated to form the time embedding vectors. Thanks for pointing this out, we will add more descriptive details to clarify it in the revised manuscript.
>
> **3. To your question about** “Moreover, the explanation about the structure of the dot model itself is somewhat confusing. It says that the pretrained model is fixed, two extra layers are added, and then LoRA is applied? This was somewhat confusing, and after seeing Fig. 3, I guessed that LoRA is applied to the pretrained part. Articulating the description would benefit the readers”.
>
> **Our responses are:** Thanks a lot for your suggestion. Yes, for the Dot model, LoRA is only applied to the pre-trained layers shared from the Dash model (fixed). The newly added two layers and LoRA are trained jointly. We will revise the corresponding description about the structure to improve its clarity.

---

### Official Review · Reviewer_vMF4 · 2025-03-17

**Overall Recommendation:** 3

**Summary:**

The paper introduces Morse, a framework for accelerating diffusion model sampling by training a lightweight model (Dot) to emulate a slower pretrained diffusion model (Dash). Dot leverages additional inputs, such as sampling history from earlier timesteps, to improve efficiency. Its architecture mirrors the original diffusion model but incorporates LoRA and additional downsampling/upsampling layers at the beginning and end. Only the newly added layers are fine-tuned, while the original layers operate at a reduced resolution, significantly lowering latency. By combining fast Dot models with slower Dash models during inference, Morse achieves a better quality-speed tradeoff, with 1.5–3.5× speedups depending on the setting. The authors evaluate Morse across various image generation benchmarks, demonstrating considerable improvements in efficiency without little degradation to output quality.

**Claims And Evidence:**

Yes, the claims made in the paper are well substantiated. And sufficient ablation studies are performed that showcase the impact of each design decision.

**Essential References Not Discussed:**

Among the relevant prior works related to accelerating diffusion model sampling, the line of work regarding optimizing timestep schedules seems to be missing, e.g. [1, 2, 3, 4].

[1] Watson, Daniel, et al. "Learning to efficiently sample from diffusion probabilistic models." arXiv preprint arXiv:2106.03802 (2021).
[2] Watson, Daniel, et al. "Learning fast samplers for diffusion models by differentiating through sample quality." International Conference on Learning Representations. 2021.
[3] Xue, Shuchen, et al. "Accelerating diffusion sampling with optimized time steps." Proceedings of the IEEE/CVF Conference on Computer Vision and Pattern Recognition. 2024.
[4] Sabour, Amirmojtaba, Sanja Fidler, and Karsten Kreis. "Align your steps: Optimizing sampling schedules in diffusion models." arXiv preprint arXiv:2404.14507 (2024).

**Experimental Designs Or Analyses:**

The experiments are generally sound. However, since this paper proposes an efficiency technique for accelerating diffusion inference, it would have been preferable to compare against other established acceleration methods, such as feature reuse, timestep schedule optimization, and quantization. Some of these approaches are training-free and do not require an additional model, making a comparison particularly relevant. A discussion of how the proposed method compares in terms of tradeoffs—such as speed, memory efficiency, and ease of adoption—would strengthen the evaluation.

**Methods And Evaluation Criteria:**

The method is very simple and easy to understand. The benchmarking suite includes a good amount of baseline models and datasets, and the evaluation criteria make sense.

**Other Comments Or Suggestions:**

No comments.

**Other Strengths And Weaknesses:**

**Strengths**:
* The paper is well written and nicely structured.
* The proposed method is simple and easy to understand.

**Weaknesses**:
* No additional efficient diffusion model baselines were compared against. Adding comparisons to prominent work in the field, such as DeepCache, and discussing the pros/cons of such approaches compared to Morse would be helpful. (Addressed in the rebuttal)

**Questions For Authors:**

.

**Relation To Broader Scientific Literature:**

The slow sampling speed of diffusion- and flow-based generative models is a well-known limitation, and a significant amount of research has focused on addressing this issue.

For few-step sampling (1–4 steps), distillation-based approaches are essential for achieving acceptable quality. However, these methods require training an additional student model, which comes with substantial computational overhead.

For slightly higher NFE regimes (10–30 steps), a variety of techniques have been proposed, including improved ODE/SDE solvers, optimized sampling schedules, feature reuse across timesteps and model layers, model quantization, etc. The approach in this paper falls within this setting—while it improves sampling in the few-step regime, the sample quality still degrades significantly at very low NFE.
Among these methods, training-free approaches tend to be more impactful due to their ease of adoption, as they do not require additional model training. Comparing this method against such alternatives would help clarify its tradeoffs in terms of efficiency and practical usability.

**Theoretical Claims:**

No major theoretical claims were made. This papers contributions is primarily an efficiency technique applicable to diffusion models.

---

> ### Author Rebuttal · Authors · 2025-03-31
>
> Thank you for recognizing our work and constructive comments.
>
> **1. To your main concern** about the comparison of Morse with existing methods for accelerating diffusion inference, **our responses include 3 parts**:
>
> **Part 1: Comparison with Feature Reuse.** Following the suggestion by you and Reviewer cEcd, we compare Morse with DeepCache (its code is public): **1)** Table A shows their results on Stable Diffusion tested with the same settings as in our paper. We can see **a)** for acceleration, DeepCache is lossy in generation quality due to reusing most of the features at step $t$ for N-1 following denoising steps, yet Morse is lossless; **b)** Morse consistently gets better throughput (e.g., 4.67 vs 2.33 images/second) and FID (8.60 vs 10.62) than DeepCache; **2)** At [this linker](https://anonymous.4open.science/r/Morse-Reviewer-vMF4), we provide more results to show the superiority of Morse.
>
> **Table A**: Results on MS-COCO(512×512) with Stable Diffusion (SD) v1.4 and DDIM solver
> |Method|FID↓|Throughput↑(images/second on a 4090 GPU)|
> |-|-|-|
> |SD (50 steps)|8.22|1.07|
> |+DeepCache (N=2, N: the cache updating at every N steps)|9.31|1.62|
> |+DeepCache (N=3)|9.55|1.94|
> |+DeepCache (N=5)|10.62|2.33|
> |+Morse (LSD=50, LSD: the Latency per Step of the baseline SD)|**8.15**|1.12|
> |SD (20 steps)|8.70|2.45|
> |+Morse (LSD=20)|8.29|2.43|
> |SD (10 steps)|10.65|4.62|
> |+Morse (LSD=10)|8.60|**4.67**|
>
> **Part 2: Comparison with Timestep Schedule Optimization.** Thanks for pointing out [1-4] that are closely related to our work. Generally, [2] presents a fast solver GGDM (a contemporary work is DPM-Solver, 1 of 5 samplers tested in our paper), and [1,3,4]  design different strategies to choose optimal time steps when giving a small number of sampling steps (e.g., 10) and a solver, and only the code of [3] is public. As [1] uses a log likelihoods metric instead of popular metrics like FID for evaluation, comparing Morse with it is not applicable. For fair comparisons with [2-4], we collect their results and ours under the same settings: **1)** Table B&C compare Morse with [2]. We can see, under the same steps-budget, **a)** [2] is mostly worse than DPM-Solver, but Morse can losslessly accelerate DPM-Solver; **b)** [2] is better than DDIM, but DDIM+Morse outperforms [2]; **2)** Table D&E compare Morse with [3-4]. We can see, when using the same solver and steps-budget, **a)** Morse is superior to [3-4]; **b)** as jump sampling is our basic component, Morse can readily accelerate this line of works (as validated on [3] in Table D).
>
> **Table B**: Results on CIFAR-10(32x32)
> |Method\LSDs|5|10|15|20|25|
> |-|-|-|-|-|-|
> |[2]|13.77|8.23|6.12|4.72|4.25|
> |DPM-Solver|268.14|6.26|4.13|3.66|3.50|
> |DPM-Solver+Morse|**13.39**|**3.93**|**3.45**|**3.45**|**3.44**|
>
> **Table C**: Results on ImageNet(64x64)
> |Method\LSDs|5|10|15|20|25|
> |-|-|-|-|-|-|
> |[2]|55.1|37.3|24.7|20.7|18.4|
> |DDIM|147.4|39.4|28.7|22.2|20.0|
> |DDIM+Morse|**46.6**|**25.8**|**21.5**|**19.2**|**18.1**|
>
> **Table D**: Results on CIFAR-10(32x32)
> |Method\LSDs|5|10|12|15|
> |-|-|-|-|-|
> |DPM-Solver++(SDE)|29.22|4.03|3.45|3.17|
> |+[3]|12.91|3.51|3.24|3.15|
> |+Morse|5.73|2.91|2.80|2.75|
> |+[3] and Morse|**4.77**|**2.89**|**2.79**|**2.74**|
> |DPM-Solver(SDE)|191.46|4.72|3.83|3.77|
> |+Morse|8.65|3.41|3.22|3.06|
>
> **Table E**: Results on ImageNet(64x64)
> |Method\LSDs|5|10|15|20|25|
> |-|-|-|-|-|-|
> |DDIM|145.0|42.5|30.3|26.6|24.8|
> |+[4]|50.4|29.2|24.2|22.3|21.4|
> |+Morse|**46.6**|**25.8**|**21.5**|**19.2**|**18.1**|
>
> **Part 3: Comparison with Quantization.** In Table F, we compare Morse with Q-Diffusion (a leading open-source diffusion quantization method). We can see: **a)** Q-Diffusion is lossy due to the low-precision representation, but our Morse is lossless; **b)** Morse can also accelerate the quantized diffusion model, showing an average speedup of 1.9×.
>
> **Table F**: Results on CIFAR-10 with DDIM
> |Method\LSDs|5|10|20|50|100|
> |-|-|-|-|-|-|
> |Baseline|41.9|13.7|7.0|4.8|4.3|
> |+Morse|**13.6**|**6.6**|**5.1**|**4.4**|**4.1**|
> |Q-Diffusion (4-bit weight, 8-bit activation)|43.7|14.2|7.6|5.6|5.1|
> |+Morse|14.6|8.0|6.5|5.3|4.8|
>
> **2. To your concern** about a discussion of tradeoffs in terms of method efficiency and usability,
>
> W.r.t. the above experiments, it can be concluded that: **1)** Morse is lossless thanks to its simple dual-model design with a small number of learnable parameters. Training-free methods such as feature reuse and quantization are lossy, but Morse can improve them; **2)** As jump sampling (JS) is our basic component (note Morse cannot speedup 1-step model as JS is not applicable), Morse can accelerate timestep schedule optimization methods[1-4], similar to the solvers/distillation-based methods tested in our paper; **3)** So, in practice, combing Morse with them can provide a promising way to attain more aggressive speedup ratios under bearable degeneration of image quality.
>
> **3. Regarding more experiments and discussions**, please see our responses to Reviewer cEcd/dKeM.

---

> > ### Comment · Reviewer_vMF4 · 2025-04-04
> >
> > Thank you for the detailed response. I appreciate the additional comparisons between Morse and prior acceleration methods, and the results are quite promising. As my main concern has been sufficiently addressed, I will update my score.

---

> > > ### Author Response · Authors · 2025-04-04
> > >
> > > We are glad to see that you are satisfied with our rebuttal and have increased your score. We will add these additional experiments and discussions into our final paper, improving the paper quality.
> > >
> > > Thanks again for your constructive comments, time and patience.

---

### Decision · Program_Chairs · 2025-05-01

**Decision:**

Accept (poster)

**Comment:**

This paper proposes a fast sampling approach for diffusion models. The idea is to train a lightweight model, called Dot, that is finetuned to take the sample history as input and improve the slow model's output. Reviewers acknowledge that the idea is novel, and results are sound (after rebuttal). Although there are more to be desired in terms of its simplicity, the AC believes that it is a reasonable contribution to the diffusion modeling field and would lean towards accepting.